# In Defense of the Unitary Scalarization for Deep Multi-Task Learning

**Vitaly Kurin**[*]
University of Oxford
vitaly.kurin@cs.ox.ac.uk

**Alessandro De Palma**[*]
University of Oxford
adepalma@robots.ox.ac.uk

**Ilya Kostrikov**
University of California, Berkeley
New York University

**Shimon Whiteson**
University of Oxford

**M. Pawan Kumar**
University of Oxford

## Abstract

Recent multi-task learning research argues against *unitary scalarization*, where training simply minimizes the sum of the task losses. Several ad-hoc multi-task optimization algorithms have instead been proposed, inspired by various hypotheses about what makes multi-task settings difficult. The majority of these optimizers require per-task gradients, and introduce significant memory, runtime, and implementation overhead. We show that unitary scalarization, coupled with standard regularization and stabilization techniques from single-task learning, matches or improves upon the performance of complex multi-task optimizers in popular supervised and reinforcement learning settings. We then present an analysis suggesting that many specialized multi-task optimizers can be partly interpreted as forms of regularization, potentially explaining our surprising results. We believe our results call for a critical reevaluation of recent research in the area.

## 1 Introduction

Multi-Task Learning (MTL) [5] exploits similarities between tasks to yield models that are more accurate, generalize better and require less training data. Owing to the success of MTL on traditional machine learning models [3, 16, 22] and of deep single-task learning across a variety of domains, a growing body of research has focused on deep MTL. The most straightforward way to train a neural network for multiple tasks at once is to minimize the sum of per-task losses. Adopting terminology from multi-objective optimization, we call this approach *unitary scalarization*.

While some work shows that multi-task networks trained via unitary scalarization exhibit superior performance to independent per-task models [29, 35], others suggest the opposite [30, 54, 58]. As a result, many explanations for the difficulty of MTL have been proposed, each motivating a new Specialized Multi-Task Optimizer (SMTO) [11, 42, 54, 62, 66]. These works typically claim that the proposed SMTO outperforms unitary scalarization, in addition to relevant prior work. However, SMTOs usually require access to per-task gradients either with respect to the shared parameters, or to the shared representation. Therefore, their reported performance gain comes at significant computation and memory cost, the overhead scaling linearly with the number of tasks. By contrast, unitary scalarization requires only the average of the gradients across tasks, which can be computed via a single backpropagation.

Existing SMTOs were introduced to solve challenges related to the optimization of the deep MTL problem. We instead postulate that the reported weakness of unitary scalarization is linked to experimental variability or to a lack of regularization, leading to the following contributions:

---

[*]Equal contribution.

36th Conference on Neural Information Processing Systems (NeurIPS 2022).

- A comprehensive experimental evaluation (§4) of recent SMTOs on popular multi-task benchmarks, showing that no SMTO consistently outperforms unitary scalarization in spite of the added complexity and overhead. In particular, either the differences between unitary scalarization and SMTOs are not statistically significant, or they can be bridged by standard regularization and stabilization techniques from the single-task literature. Our reinforcement learning (RL) experiments include optimizers previously applied only to supervised learning.

- An empirical and technical analysis of the considered SMTOs, suggesting that they reduce overfitting on the multi-task problem and hence act as regularizers (§5). We conduct an ablation study and provide a collection of novel and existing technical results that support this hypothesis.

- Code to reproduce the experiments, including a unified PyTorch [50] implementation of the considered SMTOs, is available at `https://github.com/yobibyte/unitary-scalarization-dmtl`.

We believe that our results suggest that the considered SMTOs can be often replaced by less expensive techniques. We hope that these surprising results stimulate the search for a deeper understanding of MTL.

## 2 Related Work

Before diving into details of specific SMTOs in Section 5, we provide a high-level overview of the deep MTL research. Seminal work in MTL includes *hard parameter sharing* [6]: sharing neural network parameters between all tasks with, possibly, a separate part of the model for each task. Hard parameter sharing is still the major MTL approach adopted in natural language processing [9, 12], computer vision [46], and speech recognition [53]. In this work, we implicitly assume that each parameter update employs information from all tasks. However, not all works satisfy this assumption, either due to a large number of tasks [4, 36], or simply as an implementation decision [25, 37]. In this setting, MTL resembles other problems dealing with multiple tasks, i.e., continual [32], curriculum [47], and meta-learning [24], which are not the focus of this work.

Many works strive to improve the performance of deep multi-task models. One line of research hypothesizes that conflicting per-task gradient directions lead to suboptimal models, and focuses on explicitly removing such conflicts [11, 28, 41, 42, 62, 66]. Some authors postulate that loss imbalances across tasks hinder learning, proposing loss reweighting methods [10, 30, 40]. Sener and Koltun [54] and Navon et al. [48] propose that tasks compete for model capacity and interpret MTL as multi-objective optimization in order to cope with inter-task competition. Here, we focus on algorithms that explicitly rely on per-task gradients to try to outperform unitary scalarization (§5). Research on multi-task architectures [19, 46] or MTL algorithms exclusively motivated by deterministic loss reweighting [18, 30, 43] are orthogonal to our work. Both topics are investigated by a recent survey on pixel-level multi-task computer vision problems [61], which found that the minimization of tuned weighted sums of losses (scalarizations) is empirically competitive with deterministic loss reweighting and MGDA in the considered settings. These results are extended to popular SMTOs by a critical review from Xin et al. [63], concurrent to our work, which argues that the optimization and generalization performance of SMTOs can be matched by tuning scalarization coefficients. Our work reaches a similar conclusion, demonstrating that unitary scalarization performs on par with SMTOs when coupled with standard and inexpensive regularization or stabilization techniques. In other words, Xin et al. [63] provide complementary support for the link between SMTOs and regularization by showing that tuning scalarization weights positively affects generalization.

In addition to the common supervised settings, we also consider multi-task RL, whose research can be grouped into three categories: the first adds auxiliary tasks providing additional inductive biases to speed up learning [27] on a target task. The second, based on policy distillation, uses per-task teacher models to provide labels for a multi-task model or per-task policies as regularizers [49, 51, 57]. The third directly learns a shared policy [29], possibly via an SMTO [66]. We focus on the third category, whose literature reports varying performance for unitary scalarization (better [29] or worse [66] than per-task models), indicating confounding factors in evaluation pipelines and further motivating our work. PopArt [23, 60] performs scale-invariant value function updates in order to address differences in returns across environments, showing improvements in the multi-task setting while still using unitary scalarization. PopArt does not require per-task gradients but introduces additional hyperparameters. In our work, we address the differences in rewards by normalizing them at the replay buffer level. However, we believe both unitary scalarization and SMTOs might equally benefit from PopArt.

# 3 Multi-Task Learning Optimizers

We will now describe the deep MTL training problem and popular algorithms employed for its solution. Let $(X, Y) \in \mathbb{R}^{d \times n} \times \mathbb{R}^{o \times n}$ be the training set, composed of $n$ $d$-dimensional points and $o$-dimensional labels. In addition, $\mathcal{L}_i : \mathbb{R}^{o \times n} \times \mathbb{R}^{o \times n} \to \mathbb{R}$ denotes the loss for the $i$-th task, $\boldsymbol{\theta} \in \mathbb{R}^S$ the parameter space, $\mathcal{T} := \{1, \ldots, m\}$ the set of $m$ tasks. The goal of MTL is to learn a single (generally task-aware) parametrized model $f : \mathbb{R}^S \times \mathbb{R}^{d \times n} \times \mathcal{T} \to \mathbb{R}^{o \times n}$ that performs well on all tasks $\mathcal{T}$. The parameter space is often split into a set of shared parameters across tasks (generally the majority of the architecture), denoted $\boldsymbol{\theta}_{\parallel}$, and (possibly empty) task-specific parameters, denoted $\boldsymbol{\theta}_{\perp}$, so that $\boldsymbol{\theta} := [\boldsymbol{\theta}_{\parallel}, \boldsymbol{\theta}_{\perp}]^T$. In this context, the model $f$ often takes on an encoder-decoder architecture, where the encoder $g$ learns a shared representation across tasks, and the decoders $h_i$ are task-specific predictive heads: $f(\boldsymbol{\theta}, X, i) = h_i(g(\boldsymbol{\theta}_{\parallel}, X), \boldsymbol{\theta}_{\perp})$. In this case, we denote by $\mathbf{z} = g(\boldsymbol{\theta}_{\parallel}, X) \in \mathbb{R}^{r \times n}$ the $r$-dimensional shared representation of $X$.

The training problem for MTL is typically formulated as the sum of the per-task losses [11, 54, 66]:

$$\min_{\boldsymbol{\theta}} \left[ \ \mathcal{L}^{\mathrm{MT}}(\boldsymbol{\theta}) := \sum_{i \in \mathcal{T}} \mathcal{L}_i(f(\boldsymbol{\theta}, X, i), Y) \ \right]. \tag{1}$$

**Unitary Scalarization**  The obvious way to minimize the multi-task training objective in equation (1) is to rely on a standard gradient-based algorithm. While, for simplicity, we focus on standard gradient descent rather than mini-batch stochastic gradient descent, the notation can be adapted by replacing the dataset size $n$ by the mini-batch size $b$. Equation (1) corresponds to a linear scalarization with unitary weights under a multi-objective interpretation of MTL; hence, we call the direct application of gradient descent on equation (1) *unitary scalarization*. For vanilla gradient descent, this corresponds to taking a step in the opposite direction as the one given by the sum of per-task gradients: $\nabla_{\boldsymbol{\theta}} \mathcal{L}^{\mathrm{MT}} = \sum_{i \in \mathcal{T}} \nabla_{\boldsymbol{\theta}} \mathcal{L}_i$. Per-task gradients are not required, as it suffices to directly compute the gradient of the sum $\mathcal{L}^{\mathrm{MT}}$. Hence, when relying on deep learning frameworks based on reverse-mode differentiation, such as PyTorch [50], the backward pass is performed once per iteration (rather than $m$ times). Furthermore, the memory cost is a factor $m$ less than most SMTOs, which require access to each $\nabla_{\boldsymbol{\theta}} \mathcal{L}_i$. As a consequence, unitary scalarization is simple, fast, and memory efficient. Our experiments demonstrate that, when possibly coupled with single-task regularization such as early stopping, $\ell_2$ penalty or dropout layers [56], this simple optimizer is strongly competitive with SMTOs.

**MGDA**  Sener and Koltun [54] point out that equation (1) can be cast as a multi-objective optimization problem with the following objective: $\boldsymbol{\mathcal{L}}^{\mathrm{MT}}(\boldsymbol{\theta}) := [\mathcal{L}_1(\boldsymbol{\theta}), \ldots, \mathcal{L}_m(\boldsymbol{\theta})]^T$. A commonly employed solution concept in multi-objective optimization is Pareto optimality. A point $\boldsymbol{\theta}^*$ is called Pareto-optimal if, for any another point $\boldsymbol{\theta}^\dagger$ such that $\exists i \in \mathcal{T} : \mathcal{L}_i(\boldsymbol{\theta}^\dagger) < \mathcal{L}_i(\boldsymbol{\theta}^*)$, then $\exists j \in \mathcal{T} : \mathcal{L}_j(\boldsymbol{\theta}^\dagger) > \mathcal{L}_j(\boldsymbol{\theta}^*)$. A necessary condition for Pareto optimality at a point is Pareto stationarity, defined as the lack of a shared descent direction across all losses at that point. Sener and Koltun [54] rely on Multiple-Gradient Descent Algorithm (MGDA) [14] to reach a Pareto-stationary point for shared parameters $\boldsymbol{\theta}_{\parallel}$. Intuitively, MGDA proceeds by repeatedly stepping in a shared descent direction [14, 17], which can be found by solving the following optimization problem:

$$\min_{\mathbf{g}, \epsilon} \left[ \epsilon + 1/2 \, \|\mathbf{g}\|_2^2 \right] \quad \text{s.t.} \ \nabla_{\boldsymbol{\theta}_{\parallel}} \mathcal{L}_i^T \mathbf{g} \leq \epsilon \quad \forall \, i \in \mathcal{T}, \tag{2}$$

whose dual takes the following form (corresponding to the formulation from Désidéri [14]):

$$\max_{\boldsymbol{\alpha} \geq 0} -1/2 \, \|\mathbf{g}\|_2^2 \quad \text{s.t.} \ \sum_i \alpha_i \nabla_{\boldsymbol{\theta}_{\parallel}} \mathcal{L}_i = -\mathbf{g}, \quad \sum_{i \in \mathcal{T}} \alpha_i = 1. \tag{3}$$

In other words, MGDA takes a step in a direction $\mathbf{g}$ given by the negative convex combination of per-task gradients, whose coefficients are given by solving equation (3). In practice, per-task gradients are rescaled before applying MGDA: the original authors' implementation [54] relies on $\nabla_{\boldsymbol{\theta}_{\parallel}} \mathcal{L}_i \leftarrow \nabla_{\boldsymbol{\theta}_{\parallel}} \mathcal{L}_i / \|\nabla_{\boldsymbol{\theta}_{\parallel}} \mathcal{L}_i\| \mathcal{L}_i(\boldsymbol{\theta})$. The convergence of MGDA to a Pareto-stationary point is still guaranteed after normalization [14].

**IMTL**  Impartial Multi-Task Learning (IMTL) [42] is presented as an SMTO that is not biased against any single task. It is composed of two complementary algorithmic blocks: IMTL-L, acting on task losses, and IMTL-G, acting on per-task gradients. IMTL-G follows the intuition that a multi-task optimizer should proceed along a direction $\mathbf{g} = -\sum_i \alpha_i \nabla_{\boldsymbol{\theta}_{\parallel}} \mathcal{L}_i$ that equally represents per-task

gradients. This is formulated analytically by requiring that the cosine similarity between $\mathbf{g}$ and each $\nabla_{\boldsymbol{\theta}_\|}\mathcal{L}_i$ be the same. To prevent the resulting problem from being underdetermined, Liu et al. [42] add the constraint $\sum_{i\in\mathcal{T}}\alpha_i = 1$, resulting in a problem that admits a closed-form solution for $\mathbf{g}$:

$$\mathbf{g}^T\frac{\nabla_{\boldsymbol{\theta}_\|}\mathcal{L}_1}{\left\|\nabla_{\boldsymbol{\theta}_\|}\mathcal{L}_1\right\|} = \mathbf{g}^T\frac{\nabla_{\boldsymbol{\theta}_\|}\mathcal{L}_i}{\left\|\nabla_{\boldsymbol{\theta}_\|}\mathcal{L}_i\right\|}\ \ \forall\,i\in\mathcal{T}\setminus\{1\},\quad \mathbf{g} = -\sum_i\alpha_i\nabla_{\boldsymbol{\theta}_\|}\mathcal{L}_i,\quad \sum_{i\in\mathcal{T}}\alpha_i = 1.\quad(4)$$

IMTL-L, instead, aims to reweight task losses so that they are all constant over time, and equal to 1. In order to limit oscillations of the scaling factors, the authors propose to learn them jointly with the network by minimizing a common objective via gradient descent. In particular, given $s_i \in \mathbb{R}\ \forall i\in\mathcal{T}$, Liu et al. [42] derive the following form for the joint minimization problem: $\min_{\mathbf{s},\boldsymbol{\theta}}\left[\sum_i\left(e^{s_i}\mathcal{L}_i(\boldsymbol{\theta}) - s_i\right)\right]$. As proved by Liu et al. [42], IMTL-L only has a rescaling effect on the update direction of IMTL-G. Unlike IMTL-G and the other SMTOs presented in this section, IMTL-L rescaling is designed to affect the updates for task-specific parameters $\boldsymbol{\theta}_\perp$ as well.

**PCGrad** Let us write $\cos(\mathbf{x}, \mathbf{z})$ for the cosine similarity between vectors $\mathbf{x}$ and $\mathbf{z}$. Yu et al. [66] postulate that multi-task convergence is severely slowed down if the following three conditions (named the *tragic triad*) hold at once: (i) conflicting gradient directions: $\cos(\nabla_{\boldsymbol{\theta}_\|}\mathcal{L}_i, \nabla_{\boldsymbol{\theta}_\|}\mathcal{L}_j) < 0$ for some $i, j \in \mathcal{T}$; (ii) differing gradient magnitudes: $\left\|\nabla_{\boldsymbol{\theta}_\|}\mathcal{L}_i\right\| \gg \left\|\nabla_{\boldsymbol{\theta}_\|}\mathcal{L}_j\right\|$ for some $i, j \in \mathcal{T}$; and (iii) the unitary scalarization $\mathcal{L}^{\mathrm{MT}}$ has high curvature along $\nabla_{\boldsymbol{\theta}_\|}\mathcal{L}^{\mathrm{MT}}$. The PCGrad [66] SMTO is presented as a solution to the tragic triad, targeted at the first condition. Consistent with the previous sections, let us denote the update direction by $\mathbf{g}$. Furthermore, let $[\mathbf{x}]_+ := \max(\mathbf{x}, \mathbf{0})$. Given per-task gradients $\nabla_{\boldsymbol{\theta}_\|}\mathcal{L}_i$, PCGrad iteratively projects each task gradient onto the normal plane of all the gradients with which it conflicts:

$$\left[\mathbf{g}_i \leftarrow \nabla_{\boldsymbol{\theta}_\|}\mathcal{L}_i,\ \ \mathbf{g}_i \leftarrow \mathbf{g}_i + \left[\frac{-\mathbf{g}_i^T\nabla_{\boldsymbol{\theta}_\|}\mathcal{L}_j(\mathbf{x})}{\left\|\nabla_{\boldsymbol{\theta}_\|}\mathcal{L}_j\right\|^2}\right]_+ \nabla_{\boldsymbol{\theta}_\|}\mathcal{L}_j\ \ \forall j\in\mathcal{T}\setminus\{i\}\right]\forall i\in\mathcal{T},\quad \mathbf{g} = -\sum_{i\in\mathcal{T}}\mathbf{g}_i,\ (5)$$

where the iterative updates of $\mathbf{g}_i$ with respect to $\nabla_{\boldsymbol{\theta}_\|}\mathcal{L}_j$ are performed in random order.

**GradDrop** Chen et al. [11] focus on conflicting signs across task gradient entries, arguing that such conflicts lead to gradient "tug-of-wars". The GradDrop SMTO [11], presented as a solution to this problem, proposes to randomly mask per-task gradients $\nabla_{\boldsymbol{\theta}_\|}\mathcal{L}_i$ so as to minimize such conflicts. Specifically, GradDrop computes the "positive sign purity" $p_j$ for the task gradient's $j$-th entry and then masks the $j$-th entry of each per-task gradient with probability increasing with $p_j$, if the entry is negative, or decreasing with $p_j$, if the entry is positive. Let us write $\mathbf{p} := [p_1, \ldots, p_S]$, where $S$ is the dimensionality of the parameter space (see §3), $\odot$ for the Hadamard product and $\mathbb{1}_\mathbf{a}$ for the indicator vector on condition $\mathbf{a}$. Given a vector $\mathbf{u}_i$, uniformly sampled in $[\mathbf{0}, \mathbf{1}]$ at each iteration, GradDrop takes a step in the direction given by:

$$\mathbf{g} = \sum_{i\in\mathcal{T}}\left(\begin{array}{c}-\nabla_{\boldsymbol{\theta}_\|}\mathcal{L}_i\odot\mathbb{1}_{\left(\nabla_{\boldsymbol{\theta}_\|}\mathcal{L}_i > 0\right)}\odot\mathbb{1}_{(\mathbf{u}_i > \mathbf{p})}\\-\nabla_{\boldsymbol{\theta}_\|}\mathcal{L}_i\odot\mathbb{1}_{\left(\nabla_{\boldsymbol{\theta}_\|}\mathcal{L}_i < 0\right)}\odot\mathbb{1}_{(\mathbf{u}_i < \mathbf{p})}\end{array}\right),\ \text{with}\ \ \mathbf{p} = \frac{1}{2}\left(1 + \frac{\sum_{i\in\mathcal{T}}\nabla_{\boldsymbol{\theta}_\|}\mathcal{L}_i}{\sum_{i\in\mathcal{T}}\left|\nabla_{\boldsymbol{\theta}_\|}\mathcal{L}_i\right|}\right).\ (6)$$

## 4 Experimental Evaluation

Relying on a unified experimental pipeline, we present an empirical evaluation on common MTL benchmarks of unitary scalarization (§3), of the popular SMTOs presented in §3, and of the recent RLW algorithms [40] due to their similarities with PCGrad and GradDrop (see §5.2). We benchmark against the two RLW instances that showed the best average performance in the original paper: RLW with weights sampled from a Dirichlet distribution ("RLW Diri."), and RLW with weights sampled from a Normal distribution ("RLW Norm."). The goal of this section is to assess the efficacy of a popular line of previous work, focusing on a few representative or well-established optimizers. Therefore, we forego comparison with more recent SMTOs [28, 41, 48]. Nevertheless, we point out that these algorithms often lack significant enough improvements over the optimizers we consider, or may have substantial commonalities with them (see §5.2 for Nash-MTL [48], which was published concurrently to the finalization of this work). Whenever appropriate, we employ "Unit. Scal." as shorthand for unitary scalarization. We first present supervised learning experiments (§4.1), and then evaluate on a popular reinforcement learning benchmark (§4.2).

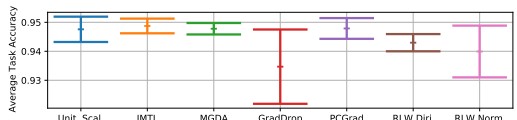 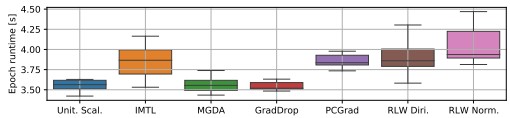

(a) Avg. task test accuracy: mean and 95% CI (10 runs).    (b) Box plots for the training time of an epoch (10 runs).

Figure 1: No algorithm outperforms unitary scalarization on the Multi-MNIST dataset.

Our experiments indicate that the performance of unitary scalarization has been consistently underestimated in the literature. By showing the variability between runs and by relying on standard regularization and stabilization techniques from the single-task literature, we demonstrate that *no SMTO consistently outperforms unitary scalarization across the considered settings*. This result holds in spite of the added complexity and computational overhead associated with most SMTOs. Furthermore, in supervised learning, most methods drive the training loss of all tasks in the proximity of the respective global minima. This suggests that the main difficulty of MTL is not associated with the optimization of its training objective, but rather to incorporating adequate regularization (cf. §5).

## 4.1 Supervised Learning

All the architectures employed in the supervised learning experiments conform to the encoder-decoder structure detailed in §3. Whenever suggested by the original authors for this context, the SMTO implementations rely on per-task gradients with respect to the last shared activation, $\nabla_{\mathbf{z}}$, rather than on the usually more expensive $\nabla_{\boldsymbol{\theta}}\mathcal{L}_i$. In particular, this is the case for MGDA, IMTL and GradDrop. See appendix B for details concerning each individual algorithm. Surprisingly, several MTL works [11, 40, 42, 66] report validation results, making it easier to overfit. Instead, following standard machine learning practice, we select a model on the validation set, and later report test metrics for all benchmarks. Validation results are also available in appendix D. Appendix C.1 reports dataset descriptions, the computational setup, hyperparameter and tuning details.

### 4.1.1 Multi-MNIST

We present results on the Multi-MNIST [54] dataset, a simple two-task supervised learning benchmark. We employ a popular architecture from previous work [54, 66] (see appendix C.1), where a single dropout layer [56] (with dropout probability $0.5$) is employed in both the encoder and the decoder. $\ell_2$ regularization did not improve validation performance and was therefore omitted. Figure 1 reports the average task test accuracy, and the training time per epoch. For each run, the test model was selected as the model with the largest average task validation accuracy across the training epochs. Appendix D presents the results of Figure 1 in tabular form, as well as the average task validation accuracy per epoch. As seen from the overlapping confidence intervals, none of the considered algorithms clearly outperforms the others. However, GradDrop displays higher experimental variability. Furthermore, Figure 7(b) shows that the sums of the task cross-entropy losses is driven nearly to zero by most methods. Finally, Figure 1(b) shows that unitary scalarization also has among the lowest training times.

### 4.1.2 CelebA

We now show results for the CelebA [44] dataset, a challenging $40$-task multi-label classification problem. We employ the same architecture as many previous studies [40, 42, 54, 66] (see appendix C.1). We tuned $\ell_2$ regularization terms $\lambda$ for all SMTOs in the following grid: $\lambda \in \{0, 10^{-4}, 10^{-3}\}$. The best validation performance was attained with $\lambda = 10^{-3}$ for unitary scalarization, IMTL and PCGrad, and with $\lambda = 10^{-4}$ for MGDA, GradDrop, and RLW. Validation performance was further stabilized

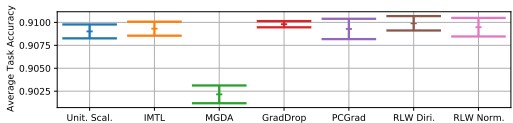 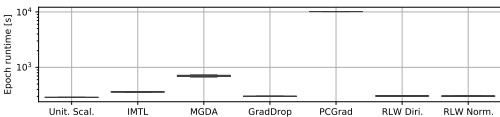

(a) Avg. task test accuracy: mean and 95% CI (3 runs).    (b) Box plots for the training time of an epoch (10 runs).

Figure 2: While SMTOs display larger runtimes, none of them outperforms the unitary scalarization on the CelebA dataset.

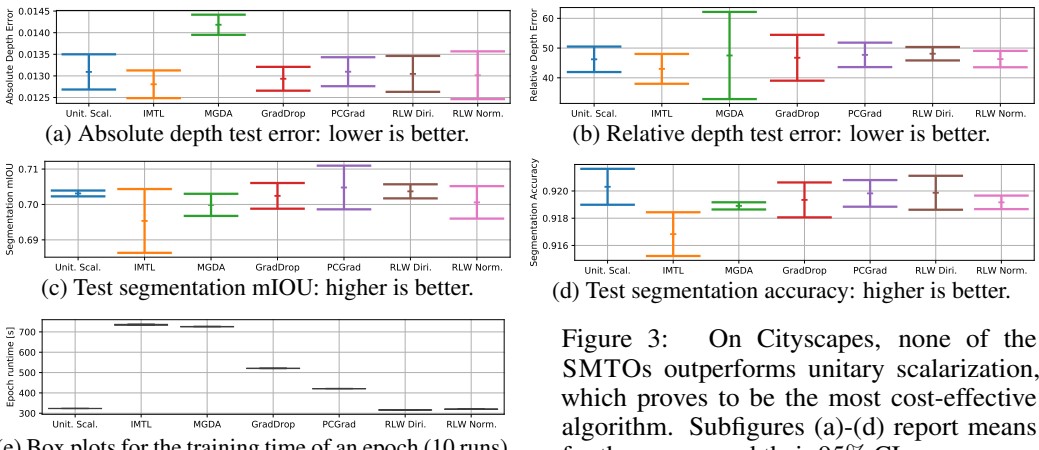

(a) Absolute depth test error: lower is better.

(b) Relative depth test error: lower is better.

(c) Test segmentation mIOU: higher is better.

(d) Test segmentation accuracy: higher is better.

(e) Box plots for the training time of an epoch (10 runs).

Figure 3: On Cityscapes, none of the SMTOs outperforms unitary scalarization, which proves to be the most cost-effective algorithm. Subfigures (a)-(d) report means for three runs, and their 95% CIs.

by the addition of several dropout layers (see Figure 5), with dropout probabilities from 0.25 to 0.5. We present an ablation study on the effect of regularization on this experiment in §5.1. Figure 10 (appendix D.2) shows that regularization improves the peak average validation performance for all the considered methods. Analogously to our Multi-MNIST results, Figure 2 plots the distribution of the training time per epoch, and the average test task accuracy. As with Multi-MNIST, the test model for each run was the one with maximal average validation task accuracy across epochs. In other words, if the peak is attained before the last epoch, we perform early stopping: as shown in Figure 8(a) in appendix D this is the case for most methods. Due to the large number of tasks, Figure 2(b) shows relatively large runtime differences across methods. PCGrad is the slowest (roughly 35 times slower than unitary scalarization). In fact, amongst the considered algorithms, it is the only one that computes per-task gradients over the parameters ($\nabla_{\boldsymbol{\theta}} \mathcal{L}_i \ \forall i \in \mathcal{T}$) at each iteration. GradDrop, MGDA and IMTL have overhead factors (compared to unitary scalarization) ranging from roughly 1.05 to 2.4 due to the relatively small size of $\mathbf{z}$ for the employed architecture. The overhead of RLW is negligible: roughly 5%. Nevertheless, due to largely overlapping confidence intervals in Figure 2(a), none of the methods consistently outperforms unitary scalarization. In fact, owing to our adoption of explicit regularization techniques (see §5.1) its average performance is superior to that reported in the literature [42, 54]. As with Multi-MNIST, Figure 9(a) demonstrates that the cross-entropy loss of each task can be driven near to its global optimum by most optimizers.

### 4.1.3 Cityscapes

In order to complement the multi-task classification experiments for Multi-MNIST and CelebA, we present results for Cityscapes [13], a dataset for semantic understanding of urban street scenes. We rely on a common encoder architecture from the literature [40, 42] (see appendix C.1), with a single dropout layer in the task-specific heads [40]. As for CelebA, unitary scalarization, IMTL, and PCGrad benefit from more regularization than the other optimizers: we employ $\lambda = 10^{-5}$ for these three algorithms, as it resulted in better validation performance on the majority of metrics, and $\lambda = 0$ for the remaining methods. Cityscapes is a heterogeneous MTL problem: it contains tasks of different types whose validation metrics cannot be averaged to perform model selection. Considering the lack of an established procedure in this context, we potentially evaluate a different model for each metric, chosen as the one with the best (maximal or minimal, depending on the metric) validation performance across epochs (we perform per-run early stopping). This procedure maximizes per-task performance, at the cost of increased inference time. If inference time is a priority, an alternative model selection procedure could rely on relative task improvement [28, 41, 48], assuming that per-metric improvements are to be weighted linearly. Nevertheless, any consistently applied model selection scheme serves the main goal of our work: evaluating all SMTOs on a fair ground. Figure 3 shows test results for two metrics per task, and the distribution of the training time per epoch. As with Multi-MNIST and CelebA, no training algorithm clearly outperforms unitary scalarization (significant overlaps across confidence intervals exist), which is again the least expensive method. In contrast with a popular hypothesis [10, 30, 42], this holds in spite of relatively large loss imbalances. In fact, the loss for the depth task is roughly 10 times smaller than that of the segmentation task: see figures 17(f)-17(g). Nevertheless, both losses are rapidly driven towards their respective global

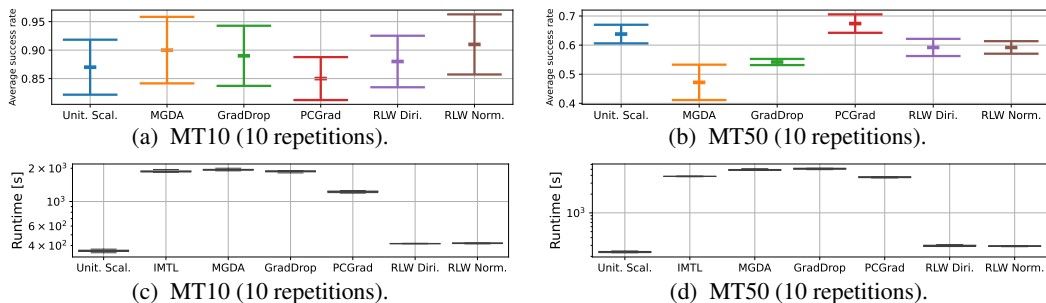

Figure 4: On Metaworld, none of the SMTOs significantly outperforms Unit. Scal., which is the least expensive method. Subfigures (a)-(b) report mean and 95% CI for the best (over the updates) average success rate. Subfigures (c)-(d) show box plots for the training time of 10,000 updates.

minima. Unlike CelebA (see Figure 2(b)), IMTL, MGDA and GradDrop are significantly slower than unitary scalarization (factors from 1.6 to 2.3), due to the relatively (compared to the parameter space) large size of $\mathbf{z}$ in the employed architecture. PCGrad, instead, appears to be less expensive (30% more than the baseline), demonstrating the benefits of working on $\nabla_{\boldsymbol{\theta}}\mathcal{L}_i$ on this model.

## 4.2 Reinforcement Learning

For RL experiments, we use Meta-World [65] and the Soft Actor-Critic [20] implementation from [55]. Unlike §4.1, the employed network architecture (see appendix C.1) is fully shared across tasks. Therefore, all SMTO implementations for these experiments rely on per-task gradients with respect to network parameters $\nabla_{\boldsymbol{\theta}}\mathcal{L}_i$ (see §5). Among the SMTOs we consider, PCGrad is the only one developed with the RL setting in mind. For fairness and completeness, we add all the other SMTOs from the supervised learning experiments, and are the first to test these optimizers in the RL setting. To stabilize learning, we increase the replay buffer size, a well known technique in single-task RL, add actor $l_2$ regularization, and modify the reward normalization employed by Sodhani et al. [55]. The unitary scalarization performance reported by Yu et al. [66] is considerably lower than that of Sodhani et al. [55], which we believe is due to the lack of reward normalization in the former. Sodhani et al. [55] keep a moving average of rewards in the environment, with a hyperparameter controlling the speed of the moving average. As we show in Figure 16, the learning algorithm is sensitive to that hyperparameter. Moreover, such normalization might make similar transitions have drastically different rewards stored in the replay buffer. To alleviate these issues, we store the raw rewards in the buffer, and normalize only when a mini-batch is sampled.

Figure 4 reports the best average success rate across the updates and the runtime for 10,000 updates. In addition to these summary statistics, reported for consistency with §4.1, the learning curves are shown in appendix E. Our MT10 (10 tasks) results in Figure 4(a) show that by stabilizing the baseline using standard RL techniques, unitary scalarization performs on par with other SMTOs, mirroring our findings in §4.1. This is in contrast with the previous literature, which reported that PCGrad outperforms unitary scalarization [55, 66]. Figure 4(b) presents results on MT50 (50 tasks): similarly to MT10, none of the SMTOs significantly outperforms unitary scalarization, with PCGrad's average being slightly above unitary scalarization. We speculate that the stochastic loss rescaling performed by PCGrad (see Proposition 3) reduces the differences in task return scales, and expect that methods like PopArt [60] would have a similar effect without requiring access to per-task gradients. While we did not tune hyperparameters for MT50 (we employed those found for MT10), it would be much easier to do that for unitary scalarization due to its lower runtime. In fact, Figure 4(d) shows that a single unitary scalarization run takes roughly 15 hours, whereas PCGrad, MGDA and GradDrop require more than a week. Similarly to MT10, actor regularization pushes the average performance of unitary scalarization higher (see in appendix E.2). Overall, as in the supervised learning setting, unitary scalarization performs comparably to SMTOs despite being simpler and less demanding in both memory and compute. IMTL was unstable on this RL benchmark and all of the runs crashed due to numerical overflow. We hence omit IMTL results from the main body of the paper and show its results in Figure 13 in appendix E, which also describes a possible explanation. We hypothesize that the instability of IMTL is due to lack of bounds on scaling coefficients. See appendix C.2 for hyperparameter settings and ablation studies.

# 5 Regularization in Specialized Multi-Task Optimizers

The empirical results presented in §4 motivate the need to carefully analyze existing SMTOs. We make an initial attempt in this direction by viewing their effects through the lens of regularization. Let us define a regularizer as a technique to reduce overfitting [15]. We first show that the SMTOs considered in §4 empirically act as regularizers via an ablation study (§5.1). We then take a closer look at their behavior, presenting technical results that support their alternative interpretation as regularizers (§5.2). Finally, §5.3 provides additional empirical backing for some of the technical results. Unless otherwise stated, we assume that MTL methods apply only to $\boldsymbol{\theta}_\parallel$ and that standard gradient-based updates are employed for tasks-specific parameters $\boldsymbol{\theta}_\perp$. We furthermore adopt the following shorthands: $\mathcal{L}_i(\boldsymbol{\theta})$ for $\mathcal{L}_i(f(\boldsymbol{\theta}, X, i), Y)$, and $\nabla_{\boldsymbol{\theta}} \mathcal{L}_i$ for $\nabla_{\boldsymbol{\theta}} \mathcal{L}_i(f(\boldsymbol{\theta}, X, i), Y)$.

## 5.1 Ablation Study

We repeat the experiment from §4.1.2 and remove explicit regularization: no dropout layers are added to the encoder-decoder architecture, and $\lambda = 0$ for all optimizers. In addition, we examine the behavior of two different $\ell_2$-regularized instances of unitary scalarization: $\lambda = 10^{-4}$ for "Unit. Scal. $\ell_2$", $\lambda = 2 \times 10^{-3}$ for "Unit. Scal. $\ell_2$+". Figure 5 shows that SMTOs behave similarly to an $\ell_2$-penalized unitary scalarization. Importantly, SMTOs delay overfitting, requiring less early stopping compared to unitary scalarization to obtain comparable performance. In other words, early stopping is sufficient for unitary scalarization to perform on par with SMTOs. Finally, overfitting is further reduced by "Unit. Scal. Reg.", which plots the regularized unitary scalarization from §4.1.2, with dropout layers and a weight decay of $\lambda = 10^{-3}$. Further results are presented in appendix D.2.

## 5.2 Technical Results

All the methods considered in §5.1 regularize more than unitary scalarization. While RLW was shown to reduce overfitting by the original authors [40, theorem 2], we now provide a collection of novel and existing technical results that potentially explain the regularizing behavior of each of the other algorithms, complementing the presentation from §3. In particular, we show that MGDA, IMTL and PCGrad have a larger convergence set than unitary scalarization, reducing the chances to land on sharp local minima [15]. Furthermore, GradDrop and PCGrad introduce significant stochasticity, which is often linked to the same effect [31, 34]. We hope these observations will steer further research.

**MGDA**  Let us denote the convex hull of a set $\mathcal{A}$ by $\mathrm{Conv}(\mathcal{A})$. We now recall a well-known property of MGDA [14] and relate it to the behavior of unitary scalarization.

**Proposition 1.** *The* MGDA SMTO *[54] converges to a superset of the convergence points of unitary scalarization. More specifically, it converges to any point $\boldsymbol{\theta}_\parallel^*$ such that:* $\mathbf{0} \in \mathrm{Conv}(\{\nabla_{\boldsymbol{\theta}_\parallel^*} \mathcal{L}_i \mid i \in \mathcal{T}\})$.

See appendix B.1 for a simple proof. As a consequence of Proposition 1, MGDA does not necessarily reach a stationary point for $\mathcal{L}^{\mathrm{MT}}$ (that is, a point for which $\sum_{i \in \mathcal{T}} \nabla_{\boldsymbol{\theta}_\parallel} \mathcal{L}_i = \mathbf{0}$) or for any of the losses $\mathcal{L}_i$ ($\nabla_{\boldsymbol{\theta}_\parallel} \mathcal{L}_i = \mathbf{0}$). For example, any point $\boldsymbol{\theta}_\parallel$ for which two per-task gradients point in opposite directions is Pareto stationary. On account of the well-known [15] relationship between under-optimizing (e.g., early stopping [7, 39]) and overfitting, proposition 1 supports the interpretation of MGDA as a regularizer for equation (1). Empirical evidence that MGDA under-optimizes is provided

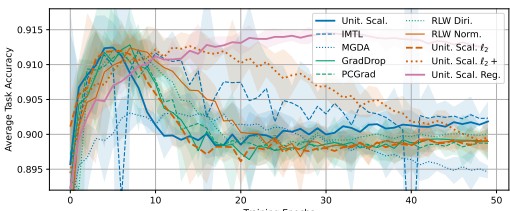

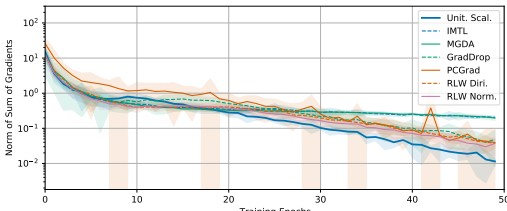

Figure 5: Mean and 95% CI (3 runs) avg. task validation accuracy over epochs on CelebA. SMTOs postpone the onset of overfitting, mirroring the effect of $\ell_2$ regularization on unitary scalarization.

Figure 6: Mean and 95% CI (3 runs) for $\left\|\sum_{i \in \mathcal{T}} \nabla_{\boldsymbol{\theta}_\parallel} \mathcal{L}_i\right\|_2$ on CelebA. MGDA and IMTL converge away from stationary points of unitary scalarization, indicating under-optimization.

in §5.3, Figure 9(a), and Figure 5, which shows over-regularization. Proposition 1 can be trivially extended to the recent Nash-MTL, which shares the same convergence set [48, Theorem 5.4].

**IMTL** We now show that aggregating per-task gradients so that their cosine similarity is the same (equation (4)) yields a constrained steepest-descent algorithm (Proposition 2). This view on the update step of IMTL leads to a novel analysis of its convergence points (corollary 1). Proofs can be found in appendix B.2. We will denote by $\text{Aff}(\mathcal{A})$ the affine hull of a set $\mathcal{A}$.

**Proposition 2.** *IMTL by Liu et al. [42] updates $\boldsymbol{\theta}_{\|}$ by taking a step in the* steepest descent direction *whose cosine similarity with per-task gradients is the same across tasks.*

**Corollary 1.** IMTL *by Liu et al. [42] converges to a superset of the Pareto-stationary points for $\boldsymbol{\theta}_{\|}$ (and hence of the convergence points of the unitary scalarization). More specifically, it converges to any point $\boldsymbol{\theta}_{\|}^*$ such that:* $\quad \mathbf{0} \in \text{Aff}\left(\left\{\nabla_{\boldsymbol{\theta}_{\|}^*} \mathcal{L}_i / \left\| \nabla_{\boldsymbol{\theta}_{\|}^*} \mathcal{L}_i \right\| \mid i \in \mathcal{T}\right\}\right).$

As seen for MGDA, corollary 1 implies that, even if the employed model $f$ has the capacity to reach the minimal loss on $\mathcal{L}^{\text{MT}}$, IMTL may stop before reaching a stationary point. Recalling the relationship between under-optimizing and overfitting [15], this supports the interpretation of IMTL as a regularizer for equation (1). This is empirically shown in §5.3, Figures 5, 9(a). In particular, unitary scalarization reaches the same average performance of IMTL but requires earlier stopping.

**PCGrad** We provide an alternative characterization of the PCGrad update rule, highlighting its stochasticity in the context of its interpretation as loss rescaling [40, 42]. See appendix B.3 for a proof.

**Proposition 3.** *PCGrad is equivalent to a dynamic, and possibly stochastic, loss rescaling for $\boldsymbol{\theta}_{\|}$. At each iteration, per-task gradients are rescaled as follows:*

$$\nabla_{\boldsymbol{\theta}_{\|}} \mathcal{L}_i \leftarrow \left(1 + \sum_{j \in \mathcal{T} \setminus \{i\}} d_{ji}\right) \nabla_{\boldsymbol{\theta}_{\|}} \mathcal{L}_i, \; d_{ji} \in \left[0, \frac{\left\| \nabla_{\boldsymbol{\theta}_{\|}} \mathcal{L}_j \right\|}{\left\| \nabla_{\boldsymbol{\theta}_{\|}} \mathcal{L}_i \right\|}\right].$$

*Furthermore, if $|\mathcal{T}| > 2$, $d_{ji}$ is a random variable, and the above range contains its support.*

The results from proposition 3 can be easily extended to GradVac [62], which generalizes PCGrad's projection onto the normal vector to arbitrary target cosine similarities between per-task gradients. When $|\mathcal{T}| > 2$, PCGrad corresponds to a stochastic loss re-weighting. As such, PCGrad bears many similarities with Random Loss Weighting (RLW) [40]. RLW proposes to sample scalarization weights from standard probability distributions at each iteration, and proves that this leads the better generalization [40, theorem 2]. Indeed, it is well-known that adding noise to stochastic gradient estimations leads the optimization towards flatter minima, and that such minima may reduce overfitting [31, 34]. In line with the main technical results by Yu et al. [66], we now restrict our focus to two-task problems, which allow for an easy description of PCGrad's convergence points. The result is largely based on [66, theorem 1]: we relax some of the assumptions and provide a proof in appendix B.3.

**Corollary 2.** *If $|\mathcal{T}| = 2$, PCGrad will stop at any point where $\cos(\nabla_{\boldsymbol{\theta}_{\|}} \mathcal{L}_1, \nabla_{\boldsymbol{\theta}_{\|}} \mathcal{L}_2) = -1$. Furthermore, if $\mathcal{L}_1$ and $\mathcal{L}_2$ are differentiable, and $\nabla_{\boldsymbol{\theta}_{\|}} \mathcal{L}^{MT}$ is L-Lipschitz with $L > 0$, PCGrad with step size $t < \frac{1}{L}$ converges to a superset of the convergence points of the unitary scalarization.*

Corollary 2 implies that, when $|\mathcal{T}| = 2$, PCGrad may under-optimize equation (1) as MGDA and IMTL. In particular, if $\cos(\nabla_{\boldsymbol{\theta}_{\|}} \mathcal{L}_1, \nabla_{\boldsymbol{\theta}_{\|}} \mathcal{L}_2) = -1$, then $\mathbf{0} \in \text{Conv}(\{\nabla_{\boldsymbol{\theta}_{\|}} \mathcal{L}_1, \nabla_{\boldsymbol{\theta}_{\|}} \mathcal{L}_2\})$ (see proposition 1). We believe that PCGrad's stochasticity and enlarged convergence set potentially explain its regularizing effect.

**GradDrop** While the motivation behind GradDrop is to avoid entry-wise gradient conflicts across tasks, the main property of the method is to drive the optimization towards "joint minima": points that are stationary for all the individual tasks at once [11, proposition 1]. In other words: $\nabla_{\boldsymbol{\theta}_{\|}} \mathcal{L}_i = \mathbf{0} \, \forall \, i \in \mathcal{T}$. While this property is desirable, we show that it holds beyond GradDrop, and independently of the gradient directions. Under strong assumptions on the model capacity, the above property would trivially hold for unitary scalarization (proposition 5, appendix B.4). Proposition 4 shows that it holds for a simple randomized version of unitary scalarization, which we name Random Grad Drop (RGD).

**Proposition 4.** *Let $\mathcal{L}^{RGD}(\boldsymbol{\theta}_{\|}) := \sum_{i \in \mathcal{T}} u_i \mathcal{L}_i(\boldsymbol{\theta}_{\|})$, where $u_i \sim Bernoulli(p) \, \forall i \in \mathcal{T}$ and $p \in (0, 1]$. The gradient $\nabla_{\boldsymbol{\theta}_{\|}} \mathcal{L}^{RGD}$ is always zero if and only if $\nabla_{\boldsymbol{\theta}_{\|}} \mathcal{L}_i = \mathbf{0} \, \forall i \in \mathcal{T}$. In other words, the result from [11, proposition 1] can be obtained without any information on the sign of per-task gradients.*

Proposition 4 (see appendix B.4 for a simple proof) shows that an inexpensive sign-independent stochastic scalarization shares GradDrop's main reported property. $\mathcal{L}^{\mathrm{RGD}}$ can be directly cast an instance of RLW, and hence as a regularization method [31, 34]. Furthermore, Figure 12 in appendix D.3 shows that the empirical results of GradDrop on CelebA [44] are closely matched by a sign-agnostic gradient masking, partly undermining the conflicting gradients assumption. We believe that the above results, along with the authors' original experiments showing that GradDrop delays overfitting on CelebA [11, figure 3], suggest that GradDrop behaves as a regularizer.

### 5.3 Under-Optimization: Empirical Study

As seen in §5.2, MGDA and IMTL might under-optimize equation (1) compared to unitary scalarization due to their larger convergence sets. In order to assess whether this is empirically the case, we estimate $\left\|\sum_{i \in \mathcal{T}} \nabla_{\boldsymbol{\theta}_{\|}} \mathcal{L}_i\right\|_2$, the norm of the unitary scalarization update on shared parameters $\boldsymbol{\theta}_{\|}$, for all optimizers throughout the unregularized CelebA experiment from §5.1. Large magnitudes for $\left\|\sum_{i \in \mathcal{T}} \nabla_{\boldsymbol{\theta}_{\|}} \mathcal{L}_i\right\|_2$ towards convergence would indicate that SMTOs steer optimization far from stationary points of unitary scalarization, resulting in under-optimization. We compute the update norm on the mini-batch loss every 100 updates, and report the per-epoch average in Figure 6. Most SMTOs have a smaller update magnitude than unitary scalarization in the first 15 epochs. However, towards convergence, SMTOs display larger $\left\|\sum_{i \in \mathcal{T}} \nabla_{\boldsymbol{\theta}_{\|}} \mathcal{L}_i\right\|_2$ compared to unitary scalarization. In particular, IMTL and MGDA have the largest norm, denoting significant empirical under-optimization. The additional stochasticity of RLW, PCGrad, and GradDrop also appears to lead to larger norm values than unitary scalarization, yet to a lesser degree. Given that MGDA and IMTL incur a larger loss than unitary scalarization in later epochs (see Figure 9(a) in appendix D.2), we can conclude that they guide optimization towards regions of the parameter space that under-optimize equation (1), providing empirical support for our analysis.

## 6   Conclusions

This paper made two main contributions. First, we evaluated popular SMTOs using a single experimental pipeline, including previously unpublished results of MGDA, IMTL, RLW, and GradDrop in the RL setting. Surprisingly, our evaluation showed that none of the SMTOs consistently outperform unitary scalarization, the simplest and least expensive method. Second, in order to explain our surprising results, we postulate that SMTOs act as regularizers and present an analysis that supports our hypothesis. We believe our work calls for further reevaluation of progress in developing principled and efficient MTL algorithms.

We conclude by addressing the limitations of our work. While we covered a wide range of popular benchmarks, we do not exclude the existence of settings where unitary scalarization underperforms: discovering them is an interesting direction for future work. Furthermore, our experimental results were obtained via grid searches under limited compute resources: some of the methods might benefit from further fine-tuning. Nevertheless, we remark that fine-tuning will be easier for unitary scalarization due to its shorter runtimes. Finally, we presented the regularization hypothesis only as a partial explanation of our results: we hope it will steer further analysis and consequently improve the understanding of MTL.

### Acknowledgements

VK was funded by Samsung R&D Institute UK through the EPSRC Centre for Doctoral Training (CDT) in Autonomous Intelligent Machines and Systems (AIMS) at the University of Oxford . ADP was funded by EPSRC for the AIMS CDT, grant EP/L015987/1, and by an IBM PhD fellowship. SW has received funding from the European Research Council under the European Union's Horizon 2020 research and innovation programme (grant agreement number 637713). The experiments were made possible by a generous equipment grant from NVIDIA. We would like to thank Lin et al. [40], Sodhani et al. [55] and Sener and Koltun [54] for publicly releasing their code. The authors thank Kristian Hartikainen for helpful comments on the RL experiments. VK thanks Ryota Tomioka for useful discussions on multitask optimization.

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
