## A  Societal Impact

Due to the object of its study, our work does not have a direct societal impact. However, as any machine learning paper, it can potentially negatively effect the society through automation and loss of jobs. While it is hard to anticipate any particular risk, as any technology, if not regulated properly, it might lead to growing social and economic inequality.

On the positive side, our work might have a positive environmental impact since it advocates for simpler and more economical methods which will reduce energy consumption in data centers. Finally, simpler methods are usually easier to understand, which is beneficial in terms of explainability, an important factor for real-life applications.

## B  Supplement to the Overview of Multi-Task Optimizers

This section presents the proofs and the technical results omitted from section 5, along with a description of the use of per-task gradients with respect to the last shared activation for encoder-decoder architectures (usually less expensive than per-task gradients with respect to shared parameters).

### B.1  MGDA

**Proposition 1.** *The* MGDA SMTO *[54] converges to a superset of the convergence points of unitary scalarization. More specifically, it converges to any point $\boldsymbol{\theta}_{\parallel}^*$ such that: $\mathbf{0} \in Conv(\{\nabla_{\boldsymbol{\theta}_{\parallel}^*}\mathcal{L}_i \mid i \in \mathcal{T}\})$.*

*Proof.* As shown by Désidéri [14], equation (3) is a simplex-constrained norm-minimization problem. In other words, the argument of the minimum is the projection of $\mathbf{0}$ onto the feasible set. Therefore:

$$\mathbf{g} = \mathbf{0} \iff \mathbf{0} \in \mathrm{Conv}(\{\nabla_{\boldsymbol{\theta}_{\parallel}}\mathcal{L}_i \mid i \in \mathcal{T}\}).$$

It then suffices to point out that $\sum_{i\in\mathcal{T}} \nabla_{\boldsymbol{\theta}_{\parallel}}\mathcal{L}_i = \mathbf{0} \iff \sum_{i\in\mathcal{T}} \frac{1}{|\mathcal{T}|}\nabla_{\boldsymbol{\theta}_{\parallel}}\mathcal{L}_i = \mathbf{0} \Rightarrow \mathbf{0} \in \mathrm{Conv}(\{\nabla_{\boldsymbol{\theta}_{\parallel}}\mathcal{L}_i \mid i \in \mathcal{T}\})$ to conclude the proof. $\square$

Due to the cost of computing per-task gradients, Sener and Koltun [54] propose MGDA-UB, which replaces the gradients wrt the parameters $\nabla_{\boldsymbol{\theta}_{\parallel}}\mathcal{L}_i$ with the gradients wrt the shared activation $\nabla_{\mathbf{z}}\mathcal{L}_i$ in the computation of the coefficients of $\mathbf{g} = -\sum_i \alpha_i \nabla_{\boldsymbol{\theta}_{\parallel}}\mathcal{L}_i$. This yields an upper bound on the objective of equation (3), thus restricting the set of points the algorithm convergences to. Rather than directly relying on $\nabla_{\boldsymbol{\theta}_{\parallel}}\mathcal{L}_i$, $\mathbf{g}$ can then be obtained by computing the gradient of $\sum_{i\in\mathcal{T}}\alpha_i\mathcal{L}_i$ via reverse-mode differentiation, hence saving memory and compute.

**Corollary 3.** *The MGDA-UB* SMTO *by Sener and Koltun [54] converges to any point such that: $\mathbf{0} \in Conv(\{\nabla_{\mathbf{z}}\mathcal{L}_i \mid i \in \mathcal{T}\})$. Furthermore, if $\frac{\partial \mathbf{z}}{\partial \boldsymbol{\theta}_{\parallel}}$ is non-singular, it converges to a superset of the convergence points of the unitary scalarization.*

*Proof.* The first part of the proof proceeds as the proof of proposition 1, noting that the MGDA-UB update is associated to the following problem:

$$\begin{aligned} \max_{\boldsymbol{\alpha}} \quad & -\frac{1}{2}\|\mathbf{g}\|_2^2 \\ \text{s.t.} \quad & \sum_i \alpha_i \nabla_{\mathbf{z}}\mathcal{L}_i = -\mathbf{g}, \quad \sum_{i\in\mathcal{T}}\alpha_i = 1, \\ & \alpha_i \geq 0 \quad \forall\, i \in \mathcal{T}. \end{aligned}$$

In order to show that a stationary point of the unitary scalarization satisfies $\mathbf{0} \in \mathrm{Conv}(\{\nabla_{\mathbf{z}^*}\mathcal{L}_i \mid i \in \mathcal{T}\})$, we will assume $\frac{\partial \mathbf{z}}{\partial \boldsymbol{\theta}_{\parallel}}$ is non-singular, as done by Sener and Koltun [54, theorem 1]. Then, relying

on the chain rule, the result follows from:

$$\sum_{i\in\mathcal{T}} \nabla_{\boldsymbol{\theta}_{\|}}\mathcal{L}_i = \mathbf{0} \iff \sum_{i\in\mathcal{T}} \frac{1}{|\mathcal{T}|}\nabla_{\boldsymbol{\theta}_{\|}}\mathcal{L}_i = \mathbf{0}$$

$$\iff \sum_{i\in\mathcal{T}} \frac{\frac{\partial \mathbf{z}}{\partial \boldsymbol{\theta}_{\|}}}{|\mathcal{T}|}\nabla_{\mathbf{z}}\mathcal{L}_i = \mathbf{0}$$

$$\iff \left(\frac{\partial \mathbf{z}}{\partial \boldsymbol{\theta}_{\|}}\right)^{-1}\frac{\partial \mathbf{z}}{\partial \boldsymbol{\theta}_{\|}}\sum_{i\in\mathcal{T}} \frac{1}{|\mathcal{T}|}\nabla_{\mathbf{z}}\mathcal{L}_i = \mathbf{0}$$

$$\iff \sum_{i\in\mathcal{T}} \frac{1}{|\mathcal{T}|}\nabla_{\mathbf{z}}\mathcal{L}_i = \mathbf{0}$$

$$\Rightarrow \mathbf{0}\in\mathrm{Conv}(\{\nabla_{\mathbf{z}}\mathcal{L}_i \mid i\in\mathcal{T}\})$$

$\square$

## B.2 IMTL

**Proposition 2.** *IMTL by Liu et al. [42] updates $\boldsymbol{\theta}_{\|}$ by taking a step in the* steepest descent direction *whose* cosine similarity with per-task gradients is the same across tasks.

*Proof.* First, equation (4) solves the linear system in $\boldsymbol{\alpha} := [\alpha_1,\dots,\alpha_m]$ given by:

$$\mathbf{g}^T\left(\frac{\nabla_{\boldsymbol{\theta}_{\|}}\mathcal{L}_1}{\|\nabla_{\boldsymbol{\theta}_{\|}}\mathcal{L}_1\|} - \frac{\nabla_{\boldsymbol{\theta}_{\|}}\mathcal{L}_i}{\|\nabla_{\boldsymbol{\theta}_{\|}}\mathcal{L}_i\|}\right) = \mathbf{0}\qquad \forall\, i\in\mathcal{T}\setminus\{1\},$$

$$\mathbf{g} = -\sum_i \alpha_i\nabla_{\boldsymbol{\theta}_{\|}}\mathcal{L}_i,\quad \sum_{i\in\mathcal{T}}\alpha_i = 1,$$

which corresponds to finding a point of $\mathcal{A}' := \mathrm{Aff}(\{\nabla_{\boldsymbol{\theta}_{\|}}\mathcal{L}_i \mid i\in\mathcal{T}\})$ which is orthogonal to $\mathcal{A} := \mathrm{Aff}\left(\left\{\frac{\nabla_{\boldsymbol{\theta}_{\|}}\mathcal{L}_i}{\|\nabla_{\boldsymbol{\theta}_{\|}}\mathcal{L}_i\|}\mid i\in\mathcal{T}\right\}\right)$. To see this, it suffices to point out that any point orthogonal to $\mathcal{A}$ is also orthogonal to the vector subspace spanned by differences of vectors belonging to $\mathcal{A}$. As this subspace has $m-1$ dimensions, any vector orthogonal to $\left(\frac{\nabla_{\boldsymbol{\theta}_{\|}}\mathcal{L}_1}{\|\nabla_{\boldsymbol{\theta}_{\|}}\mathcal{L}_1\|} - \frac{\nabla_{\boldsymbol{\theta}_{\|}}\mathcal{L}_i}{\|\nabla_{\boldsymbol{\theta}_{\|}}\mathcal{L}_i\|}\right)$ for each $i\in\mathcal{T}\setminus\{1\}$ is orthogonal to the entire subspace.

Second, consider the problem of finding a point in $\mathcal{A}$ that is orthogonal to the linear subspace spanned by differences of vectors in $\mathcal{A}$. In other words, we seek the projection of $\mathbf{0}$ onto $\mathcal{A}$. Recalling the definition of $\mathcal{A}$, we can write:

$$\begin{aligned}
\max_{\boldsymbol{\alpha}}\quad & -\frac{1}{2}\|\mathbf{g}'\|_2^2 \\
\text{s.t.}\quad & \sum_i \alpha_i\frac{\nabla_{\boldsymbol{\theta}_{\|}}\mathcal{L}_i}{\|\nabla_{\boldsymbol{\theta}_{\|}}\mathcal{L}_i\|} = -\mathbf{g}',\quad \sum_i\alpha_i = 1.
\end{aligned} \tag{7}$$

The solution of equation (7) is always collinear to the solution of equation (4). In fact, if a vector $\mathbf{g}\in\mathcal{A}'$ is orthogonal to the affine subspace $\mathcal{A}$ (or to the linear subspace spanned by differences of its members), then $\gamma\mathbf{g} = \left(-\gamma\sum_i\left(\alpha_i\|\nabla_{\boldsymbol{\theta}_{\|}}\mathcal{L}_i\|\right)\frac{\nabla_{\boldsymbol{\theta}_{\|}}\mathcal{L}_i}{\|\nabla_{\boldsymbol{\theta}_{\|}}\mathcal{L}_i\|}\right)$ is orthogonal to $\mathcal{A}$ as well, and $\gamma = \frac{1}{\sum_i\left(\alpha_i\|\nabla_{\boldsymbol{\theta}_{\|}}\mathcal{L}_i\|\right)} \implies \gamma\mathbf{g}\in\mathcal{A}$.

Finally, equation (7) differs from equation (3) in two aspects: $\boldsymbol{\alpha}$ is not constrained to be non-negative (hence the convex hull is replaced by the affine hull), and the task vectors are normalized. Therefore, equation (7) is the dual of:

$$\begin{aligned}
\min_{\mathbf{g},\epsilon}\quad & \epsilon + \frac{1}{2}\|\mathbf{g}\|_2^2 \\
\text{s.t.}\quad & \frac{\nabla_{\boldsymbol{\theta}_{\|}}\mathcal{L}_i^T}{\|\nabla_{\boldsymbol{\theta}_{\|}}\mathcal{L}_i\|}\mathbf{g} = \epsilon\qquad \forall\, i\in\{1,\dots,m\}.
\end{aligned} \tag{8}$$

The proposition then follows by comparing equation (8) with equation (2), and recalling that IMTL-L only adds a scaling factor to the chosen update direction. □

**Corollary 1.** IMTL *by Liu et al. [42] converges to a superset of the Pareto-stationary points for* $\boldsymbol{\theta}_{\parallel}$ *(and hence of the convergence points of the unitary scalarization). More specifically, it converges to any point* $\boldsymbol{\theta}_{\parallel}^*$ *such that:* $\quad \mathbf{0} \in \mathit{Aff}\left(\left\{\nabla_{\boldsymbol{\theta}_{\parallel}^*}\mathcal{L}_i / \left\|\nabla_{\boldsymbol{\theta}_{\parallel}^*}\mathcal{L}_i\right\| \mid i \in \mathcal{T}\right\}\right).$

*Proof.* Inspecting equation (8), which yields a collinear point to the IMTL update, reveals that IMTL might converge to non Pareto-stationary points: due to the restrictive equality constraints, the minimizer of equation (8) might be $\mathbf{0}$ even if a descent direction exists. Furthermore, its dual, equation (7), implies that:

$$\mathbf{g} = \mathbf{0} \iff \mathbf{0} \in \mathrm{Aff}\left(\left\{\frac{\nabla_{\boldsymbol{\theta}_{\parallel}}\mathcal{L}_i}{\|\nabla_{\boldsymbol{\theta}_{\parallel}}\mathcal{L}_i\|} \mid i \in \mathcal{T}\right\}\right)$$
$$\iff \mathbf{0} \in \mathrm{Aff}\left(\{\nabla_{\boldsymbol{\theta}_{\parallel}}\mathcal{L}_i \mid i \in \mathcal{T}\}\right),$$

which, noting that $\mathrm{Conv}(\mathcal{A}) \subseteq \mathrm{Aff}(\mathcal{A})$ for any $\mathcal{A}$, concludes the proof. □

Similarly to MGDA-UB, Liu et al. [42] advocate using $\nabla_{\mathbf{z}}\mathcal{L}_i$ in place of $\nabla_{\boldsymbol{\theta}_{\parallel}}\mathcal{L}_i$ while solving equation (4), typically reducing the cost of computing the coefficients of $\mathbf{g} = -\sum_i \alpha_i \nabla_{\boldsymbol{\theta}_{\parallel}}\mathcal{L}_i$.

**Corollary 4.** *When employing the approximation of problem (4) that relies on* $\nabla_{\mathbf{z}}\mathcal{L}_i$, *IMTL by Liu et al. [42] converges to* $\mathbf{0} \in \mathit{Aff}\left(\left\{\frac{\nabla_{\mathbf{z}}\mathcal{L}_i}{\|\nabla_{\mathbf{z}}\mathcal{L}_i\|} \mid i \in \mathcal{T}\right\}\right)$. *If* $\frac{\partial \mathbf{z}}{\partial \boldsymbol{\theta}_{\parallel}}$ *is non-singular, this is a superset of of the convergence points of the unitary scalarization.*

*Proof.* Following the proof of proposition 2, the following problem yields a collinear point to the $\nabla_{\mathbf{z}}\mathcal{L}_i$-approximate IMTL update:

$$\max_{\boldsymbol{\alpha}} \quad -\frac{1}{2}\|\mathbf{g}'\|_2^2$$
$$\text{s.t.} \quad \sum_i \alpha_i \frac{\nabla_{\mathbf{z}}\mathcal{L}_i}{\|\nabla_{\mathbf{z}}\mathcal{L}_i\|} = -\mathbf{g}', \quad \sum_i \alpha_i = 1.$$

Therefore:

$$\mathbf{g} = \mathbf{0} \iff \mathbf{0} \in \mathrm{Aff}\left(\left\{\frac{\nabla_{\mathbf{z}}\mathcal{L}_i}{\|\nabla_{\mathbf{z}}\mathcal{L}_i\|} \mid i \in \mathcal{T}\right\}\right).$$

Finally, assuming $\frac{\partial \mathbf{z}}{\partial \boldsymbol{\theta}_{\parallel}}$ is non-singular, we can replicate the procedure in the proof of corollary 3 to get:

$$\sum_{i \in \mathcal{T}} \nabla_{\boldsymbol{\theta}_{\parallel}}\mathcal{L}_i = \mathbf{0} \iff \sum_{i \in \mathcal{T}} \frac{1}{|\mathcal{T}|}\nabla_{\mathbf{z}}\mathcal{L}_i = \mathbf{0}$$
$$\iff \sum_{i \in \mathcal{T}} \frac{\|\nabla_{\mathbf{z}}\mathcal{L}_i\|}{|\mathcal{T}|} \frac{\nabla_{\mathbf{z}}\mathcal{L}_i}{\|\nabla_{\mathbf{z}}\mathcal{L}_i\|} = \mathbf{0}$$
$$\iff \left(\frac{|\mathcal{T}|}{\sum_{i \in \mathcal{T}}(\|\nabla_{\mathbf{z}}\mathcal{L}_i\|)}\right)\sum_{i \in \mathcal{T}} \frac{\|\nabla_{\mathbf{z}}\mathcal{L}_i\|}{|\mathcal{T}|} \frac{\nabla_{\mathbf{z}}\mathcal{L}_i}{\|\nabla_{\mathbf{z}}\mathcal{L}_i\|} = \mathbf{0}$$
$$\Rightarrow \mathbf{0} \in \mathrm{Conv}\left(\left\{\frac{\nabla_{\mathbf{z}}\mathcal{L}_i}{\|\nabla_{\mathbf{z}}\mathcal{L}_i\|} \mid i \in \mathcal{T}\right\}\right)$$
$$\Rightarrow \mathbf{0} \in \mathrm{Aff}\left(\left\{\frac{\nabla_{\mathbf{z}}\mathcal{L}_i}{\|\nabla_{\mathbf{z}}\mathcal{L}_i\|} \mid i \in \mathcal{T}\right\}\right),$$

which shows that $\mathrm{Aff}\left(\left\{\frac{\nabla_{\mathbf{z}}\mathcal{L}_i}{\|\nabla_{\mathbf{z}}\mathcal{L}_i\|} \mid i \in \mathcal{T}\right\}\right)$ contains the convergence points of the unitary scalarization. □

### B.3 PCGrad

**Proposition 3.** *PCGrad is equivalent to a dynamic, and possibly stochastic, loss rescaling for $\boldsymbol{\theta}_\|$. At each iteration, per-task gradients are rescaled as follows:*

$$\nabla_{\boldsymbol{\theta}_\|}\mathcal{L}_i \leftarrow \left(1 + \sum_{j \in \mathcal{T}\backslash\{i\}} d_{ji}\right) \nabla_{\boldsymbol{\theta}_\|}\mathcal{L}_i, \ d_{ji} \in \left[0, \frac{\left\|\nabla_{\boldsymbol{\theta}_\|}\mathcal{L}_j\right\|}{\left\|\nabla_{\boldsymbol{\theta}_\|}\mathcal{L}_i\right\|}\right].$$

*Furthermore, if $|\mathcal{T}| > 2$, $d_{ji}$ is a random variable, and the above range contains its support.*

*Proof.* We start by pointing out that:

$$
\left[\frac{-\mathbf{g}_i^T \nabla_{\boldsymbol{\theta}_\|}\mathcal{L}_j(\mathbf{x})}{\left\|\nabla_{\boldsymbol{\theta}_\|}\mathcal{L}_j\right\|^2}\right]_+ = \left[\frac{-\mathbf{g}_i^T \nabla_{\boldsymbol{\theta}_\|}\mathcal{L}_j(\mathbf{x})}{\left\|\nabla_{\boldsymbol{\theta}_\|}\mathcal{L}_j\right\|}\right]_+ \frac{1}{\left\|\nabla_{\boldsymbol{\theta}_\|}\mathcal{L}_j\right\|}
$$

$$
= \left[-\cos(\mathbf{g}_i, \nabla_{\boldsymbol{\theta}_\|}\mathcal{L}_j) \left\|\mathbf{g}_i\right\|\right]_+ \frac{1}{\left\|\nabla_{\boldsymbol{\theta}_\|}\mathcal{L}_j\right\|}
$$

$$
\in \left[0, \frac{\left\|\mathbf{g}_i\right\|}{\left\|\nabla_{\boldsymbol{\theta}_\|}\mathcal{L}_j\right\|}\right].
$$

As $\mathbf{g}_i$ is obtained by iterative projections of $\nabla_{\boldsymbol{\theta}_\|}\mathcal{L}_i$ onto the normals of $\nabla_{\boldsymbol{\theta}_\|}\mathcal{L}_j \ \forall j \in \mathcal{T}\backslash\{i\}$, and the norm of a vector can only decrease or remain unvaried after projections, we can write the coefficient of each $\mathbf{g}_i$ update as:

$$
d_{ij} := \left[\frac{-\mathbf{g}_i^T \nabla_{\boldsymbol{\theta}_\|}\mathcal{L}_j(\mathbf{x})}{\left\|\nabla_{\boldsymbol{\theta}_\|}\mathcal{L}_j\right\|^2}\right]_+ \in \left[0, \frac{\left\|\nabla_{\boldsymbol{\theta}_\|}\mathcal{L}_i\right\|}{\left\|\nabla_{\boldsymbol{\theta}_\|}\mathcal{L}_j\right\|}\right], \ \ \forall i \neq j.
$$

Furthermore, if $|\mathcal{T}| > 2$ the contraction factor $\frac{\left\|\mathbf{g}_i\right\|}{\left\|\nabla_{\boldsymbol{\theta}_\|}\mathcal{L}_i\right\|}$ for the norm of $g_i$ depends on the ordering of the projections, which is stochastic by design [66]. Therefore, $d_{ij}$ a random variable whose support is contained in $\left[0, \frac{\left\|\nabla_{\boldsymbol{\theta}_\|}\mathcal{L}_i\right\|}{\left\|\nabla_{\boldsymbol{\theta}_\|}\mathcal{L}_j\right\|}\right]$. Finally, exploiting the definition of $d_{ij}$, we can re-write equation (5) as:

$$
-\mathbf{g} = \sum_{i \in \mathcal{T}} \nabla_{\boldsymbol{\theta}_\|}\mathcal{L}_i + \sum_{i \in \mathcal{T}} \sum_{j \in \mathcal{T}\backslash\{i\}} d_{ij}\nabla_{\boldsymbol{\theta}_\|}\mathcal{L}_j = \sum_{i \in \mathcal{T}} \nabla_{\boldsymbol{\theta}_\|}\mathcal{L}_i + \sum_{j \in \mathcal{T}} \sum_{i \in \mathcal{T}\backslash\{j\}} d_{ji}\nabla_{\boldsymbol{\theta}_\|}\mathcal{L}_i
$$

$$
= \sum_{j \in \mathcal{T}} \nabla_{\boldsymbol{\theta}_\|}\mathcal{L}_j + \sum_{j \in \mathcal{T}} \sum_{i \in \mathcal{T}\backslash\{j\}} d_{ji}\nabla_{\boldsymbol{\theta}_\|}\mathcal{L}_i = \sum_{j \in \mathcal{T}} \left(\sum_{i \in \mathcal{T}\backslash\{j\}} d_{ji}\nabla_{\boldsymbol{\theta}_\|}\mathcal{L}_i + \nabla_{\boldsymbol{\theta}_\|}\mathcal{L}_j\right).
$$

Introducing (and then removing, using their definition) dummy variables $d_{jj} = 1$:

$$
-\mathbf{g} = \sum_{j \in \mathcal{T}} \left(\sum_{i \in \mathcal{T}\backslash\{j\}} d_{ji}\nabla_{\boldsymbol{\theta}_\|}\mathcal{L}_i + d_{jj}\nabla_{\boldsymbol{\theta}_\|}\mathcal{L}_j\right) = \sum_{j \in \mathcal{T}} \left(\sum_{i \in \mathcal{T}} d_{ji}\nabla_{\boldsymbol{\theta}_\|}\mathcal{L}_i\right) = \sum_{i \in \mathcal{T}} \left(\sum_{j \in \mathcal{T}} d_{ji}\nabla_{\boldsymbol{\theta}_\|}\mathcal{L}_i\right)
$$

$$
= \sum_{i \in \mathcal{T}} \nabla_{\boldsymbol{\theta}_\|}\mathcal{L}_i \left(\sum_{j \in \mathcal{T}} d_{ji}\right) = \sum_{i \in \mathcal{T}} \nabla_{\boldsymbol{\theta}_\|}\mathcal{L}_i \left(1 + \sum_{j \in \mathcal{T}\backslash\{i\}} d_{ji}\right),
$$

from which the result trivially follows. $\qquad\square$

**Corollary 2.** *If $|\mathcal{T}| = 2$, PCGrad will stop at any point where $\cos(\nabla_{\boldsymbol{\theta}_\|}\mathcal{L}_1, \nabla_{\boldsymbol{\theta}_\|}\mathcal{L}_2) = -1$. Furthermore, if $\mathcal{L}_1$ and $\mathcal{L}_2$ are differentiable, and $\nabla_{\boldsymbol{\theta}_\|}\mathcal{L}^{MT}$ is L-Lipschitz with $L > 0$, PCGrad with step size $t < \frac{1}{L}$ converges to a superset of the convergence points of the unitary scalarization.*

*Proof.* Let us start from the first statement, which does not require any assumption on the loss landscape. From proposition 3, we get:

$$-\mathbf{g} = \nabla_{\boldsymbol{\theta}_\|}\mathcal{L}_1\left(1 + d_{21}\right) + \nabla_{\boldsymbol{\theta}_\|}\mathcal{L}_2\left(1 + d_{12}\right)$$

$$= \left(1 + \left[\frac{-\cos(\nabla_{\boldsymbol{\theta}_\|}\mathcal{L}_1, \nabla_{\boldsymbol{\theta}_\|}\mathcal{L}_2)\left\|\nabla_{\boldsymbol{\theta}_\|}\mathcal{L}_2\right\|}{\left\|\nabla_{\boldsymbol{\theta}_\|}\mathcal{L}_1\right\|}\right]_+\right)\nabla_{\boldsymbol{\theta}_\|}\mathcal{L}_1$$

$$+ \left(1 + \left[\frac{-\cos(\nabla_{\boldsymbol{\theta}_\|}\mathcal{L}_1, \nabla_{\boldsymbol{\theta}_\|}\mathcal{L}_2)\left\|\nabla_{\boldsymbol{\theta}_\|}\mathcal{L}_1\right\|}{\left\|\nabla_{\boldsymbol{\theta}_\|}\mathcal{L}_2\right\|}\right]_+\right)\nabla_{\boldsymbol{\theta}_\|}\mathcal{L}_2,$$

which shows that, in case of conflicting gradient directions, gradient norms are rebalanced proportionally to the angle between them. For $\cos(\nabla_{\boldsymbol{\theta}_\|}\mathcal{L}_1, \nabla_{\boldsymbol{\theta}_\|}\mathcal{L}_2) = -1$, the above evaluates to:

$$-\mathbf{g} = \left(\frac{\left\|\nabla_{\boldsymbol{\theta}_\|}\mathcal{L}_1\right\| + \left\|\nabla_{\boldsymbol{\theta}_\|}\mathcal{L}_2\right\|}{\left\|\nabla_{\boldsymbol{\theta}_\|}\mathcal{L}_1\right\|}\right)\nabla_{\boldsymbol{\theta}_\|}\mathcal{L}_1 + \left(\frac{\left\|\nabla_{\boldsymbol{\theta}_\|}\mathcal{L}_1\right\| + \left\|\nabla_{\boldsymbol{\theta}_\|}\mathcal{L}_2\right\|}{\left\|\nabla_{\boldsymbol{\theta}_\|}\mathcal{L}_2\right\|}\right)\nabla_{\boldsymbol{\theta}_\|}\mathcal{L}_2.$$

The first part of the result then follows by pointing out that, if $\cos(\nabla_{\boldsymbol{\theta}_\|}\mathcal{L}_1, \nabla_{\boldsymbol{\theta}_\|}\mathcal{L}_2) = -1$, then $\nabla_{\boldsymbol{\theta}_\|}\mathcal{L}_1 = -\nabla_{\boldsymbol{\theta}_\|}\mathcal{L}_2$, and hence $\mathbf{g} = \mathbf{0}$. We remark that a similar proof appears in [66, theorem 1 and proposition 1]. However, our derivation relaxes the author's assumptions on $\mathcal{L}^{\mathrm{MT}}$ and is therefore applicable to the training of neural networks.

Finally, given the assumptions on differentiability and smoothness, we need to prove that PCGrad converges to the stationary points of the unitary scalarization: this directly follows from [66, proposition 1]. □

## B.4 GradDrop

**Proposition 5.** *Let us assume, as often demonstrated in the single-task case [1, 45], that the multi-task network has the capacity to interpolate the data on all tasks at once: $\min_{\boldsymbol{\theta}} \mathcal{L}^{MT} = \sum_{i\in\mathcal{T}}\min_{\boldsymbol{\theta}}\mathcal{L}_i$, and that its training by gradient descent attains such global minimum. Then, if $\inf_{\boldsymbol{\theta}}\mathcal{L}_i > -\infty \,\forall\, i \in \mathcal{T}$, unitary scalarization converges to a joint minimum.*

*Proof.* It suffices to point out that if $\mathcal{L}^{\mathrm{MT}}(\boldsymbol{\theta}^*) = \sum_{i\in\mathcal{T}}\min_{\boldsymbol{\theta}}\mathcal{L}_i$, then the globally optimal loss is attained for all tasks. In other words $\mathcal{L}_i(\boldsymbol{\theta}^*) = \min_{\boldsymbol{\theta}}\mathcal{L}_i \,\forall i \in \mathcal{T}$, and hence $\nabla_{\boldsymbol{\theta}^*}\mathcal{L}_i = \mathbf{0} \,\forall\, i \in \mathcal{T}$ (joint minimum). Furthermore, running gradient descent on $\min_{\boldsymbol{\theta}}\mathcal{L}^{\mathrm{MT}}$ corresponds to the unitary scalarization (§3), which concludes the proof. □

**Proposition 4.** *Let $\mathcal{L}^{RGD}(\boldsymbol{\theta}_\|) := \sum_{i\in\mathcal{T}} u_i\mathcal{L}_i(\boldsymbol{\theta}_\|)$, where $u_i \sim Bernoulli(p) \,\forall i \in \mathcal{T}$ and $p \in (0,1]$. The gradient $\nabla_{\boldsymbol{\theta}_\|}\mathcal{L}^{RGD}$ is always zero if and only if $\nabla_{\boldsymbol{\theta}_\|}\mathcal{L}_i = \mathbf{0} \,\forall i \in \mathcal{T}$. In other words, the result from [11, proposition 1] can be obtained without any information on the sign of per-task gradients.*

Proposition 4 can be proved by adapting the proof from Chen et al. [11, proposition 1]: it suffices to replace $f(\mathcal{P})$ with the Bernoulli parameter $p$, which is non-negative by definition. In our opinion, this seriously undermines the conflicting gradient hypothesis that motivated GradDrop. For the reader's convenience, we now provide a straightforward and self-contained proof.

*Proof.* Let us start from the statement on $\nabla_{\boldsymbol{\theta}_\|}\mathcal{L}^{\mathrm{RGD}}$. If $\nabla_{\boldsymbol{\theta}_\|}\mathcal{L}_i = \mathbf{0} \,\forall i \in \mathcal{T}$, then $\nabla_{\boldsymbol{\theta}_\|}\mathcal{L}^{\mathrm{RGD}} = \mathbf{0}$ with probability one. On the other hand, if $\exists j : \nabla_{\boldsymbol{\theta}_\|}\mathcal{L}_j \neq \mathbf{0}$, then:

$$\mathbb{P}\left[\nabla_{\boldsymbol{\theta}_\|}\mathcal{L}^{\mathrm{RGD}} \neq \mathbf{0}\right] \geq \mathbb{P}\left[\nabla_{\boldsymbol{\theta}_\|}\mathcal{L}^{\mathrm{RGD}} = \nabla_{\boldsymbol{\theta}_\|}\mathcal{L}_j\right]$$
$$= p(1-p)^{m-1} > 0,$$

where the first inequality comes from the fact that $\nabla_{\boldsymbol{\theta}_\|}\mathcal{L}^{\mathrm{RGD}} = \nabla_{\boldsymbol{\theta}_\|}\mathcal{L}_j$ is only one of the many instances of a non-null $\nabla_{\boldsymbol{\theta}_\|}\mathcal{L}^{\mathrm{RGD}}$. □

Let $\mathrm{sign}(\mathbf{x})$ stand for the element-wise sign operator applied on $\mathbf{x}$. On encoder-decoder architectures, similarly to MGDA and IMTL (see appendices B.1 and B.2), the authors do not apply GradDrop on $\nabla_{\boldsymbol{\theta}_\|}\mathcal{L}_i$, but rather on a the usually less expensive $\nabla_{\mathbf{z}}\mathcal{L}_i$. In more detail, they compute the

GradDrop sign purity scores $\mathbf{p}$ from equation (6) on $\sum_{i=1}^{n} \left( \text{sign}(\mathbf{z}) \odot \nabla_{\mathbf{z}} \mathcal{L}_i \right) [i] \in \mathbb{R}^r$, and then apply equation (6) on the $\nabla_{\mathbf{z}} \mathcal{L}_i$ gradients, yielding a vector $\mathbf{g}_z \in \mathbb{R}^{n \times r}$. Then, relying on reverse-mode differentiation, the update direction in the space of the parameters $\boldsymbol{\theta}_{\|}$ is obtained via a Jacobian-vector product: $\mathbf{g} = - \left( \frac{\partial \mathbf{z}}{\partial \boldsymbol{\theta}_{\|}} \right)^T \mathbf{g}_z$. Such a computation replaces the similar $\nabla_{\boldsymbol{\theta}_{\|}} \mathcal{L}^{\text{MT}} = \left( \frac{\partial \mathbf{z}}{\partial \boldsymbol{\theta}_{\|}} \right)^T \nabla_{\mathbf{z}} \mathcal{L}^{\text{MT}}$ from the unitary scalarization.

# C  Experimental Setting, Reproducibility

We now present details concerning the experimental settings from §4, including details on the employed open-source software, dataset information, hardware specifications, and hyper-parameters.

## C.1  Supervised Learning

All the experiments were run under Ubuntu 18.04 LTS, on a single GPU per run (using two 8-GPU machines in total). Timing experiments were all run on Nvidia GeForce GTX 1080 Ti GPUs, with an Intel Xeon E5-2650 CPU. The remaining experiments were run on either Nvidia GeForce RTX 2080 Ti GPUs or Nvidia GeForce GTX 1080 Ti GPUs, respectively using an Intel Xeon Gold 6230 CPU or an Intel Xeon E5-2650 CPU.

### C.1.1  MultiMNIST

Multi-MNIST, originally introduced by Sabour et al. [52] and as modified by Sener and Koltun [54], is a simple two-task supervised learning benchmark dataset constructed by uniformly sampling MNIST [38] images, and placing one in the top-left corner, the other in the bottom-right corner. Each of the two overlaid images corresponds to a 10-class classification task. Using the above procedure, we generate the Multi-MNIST training set from the first 50000 MNIST training images, the validation set from the last 10000 training images, and the test set from the original MNIST test set. For consistency with the experimental setup of Sener and Koltun [54], we employ a modified encoder-decoder version of the LeNet architecture [38]. Specifically, the last layer is omitted from the encoder, and two fully-connected layers are employed as task-specific predictive heads. The cross-entropy loss is used for both tasks. All methods are trained for 100 epochs using Adam [33] in the stochastic gradient setting, with an initial learning rate of $\eta = 10^{-2}$ (tuned in $\eta \in \{10^{-3}, 10^{-2}, 10^{-1}\}$ and yielding the best validation results for all considered algorithms), exponentially decayed by 0.95 after each epoch, and a mini-batch size of 256.

### C.1.2  CelebA

The CelebA [44] dataset consists of $200,000$ headshots (with standard training, validation and test splits) associated with the presence or absence of 40 attributes. In the MTL literature, is commonly treated as a 40-task classification problem, each task being a binary classification problem for an attribute. As commonly done in previous work [42, 54, 66], we employ an encoder-decoder architecture where the encoder is a ResNet-18 [21] (without the final layer) with batch normalization layers [26], and the per-task decoders are linear classifiers. The cross-entropy loss is used for all tasks. The training is performed from scratch for 50 epochs using Adam, with a mini-batch size of 128 and a per-epoch exponential decay factor of 0.95. As common on this network-dataset combination [11, 40], the initial learning rate is $\eta = 10^{-3}$ for all methods except for MGDA and IMTL, for which $\eta = 5 \times 10^{-4}$ yielded a better validation performance. As done by the respective authors, for PCGrad, RLW and GradDrop we use the same learning rate as the unitary scalarization [11, 40, 66].

### C.1.3  Cityscapes

We rely on the version of the dataset pre-processed by Liu et al. [43], which consists of $2,975$ training and 500 test images and presents two tasks: semantic segmentation on 7 classes, and depth estimation. We further split the original training set into a validation set of 595 images, employed to tune hyper-parameters, and a training set of 2380 images. Consistently with recent work [40], we rely on a dilated ResNet-50 architecture pre-trained on ImageNet [64] for the encoder, and on the Atrous Spatial Pyramid Pooling [8], which internally uses batch normalization, as decoders. While more powerful encoders might lead to better performance on Cityscapes, like the SegNet [2] used in [28, 41, 48], we aim to provide a fair comparison of MTL optimizers, rather than maximize overall task performance. Cross-entropy loss is employed on the semantic segmentation task, whereas the $\ell_1$ loss is used for the depth estimation. The training is performed by using Adam with a mini-batch size of 32 for 100 epochs, with an initial step size $\eta = 5 \times 10^{-4}$ resulting in the best validation performance for all algorithms, exponentially decayed by 0.95 at each epoch.

## C.2 Reinforcement Learning

Similarly to the supervised learning experiments, we ran all the experiments under Ubuntu 18.04 LTS using one GPU per run (using six 8-GPU machines in total). Timing experiments were all run using NVIDIA GeForce RTX 2080 Ti GPUs, with an Intel Xeon Gold 6230 CPU. The main bulk of the remaining experiments was run on Nvidia GeForce RTX 2080 Ti GPUs with either Intel Xeon Gold 6230 or Intel Xeon Silver 4216. We utilised NVIDIA GeForce RTX 3080 GPUs with Intel Xeon Gold 6230 CPUs for a small fraction of experiments.

We use Meta-World's MT10/MT50 for our experiment. The benchmark consists of ten/fifty tasks in which a simulated robot manipulator has to perform various actions, e.g., pressing a button, opening a door, or pushing the block. We use Sodhani et al. [55] for most of the hyperparameters and list them in Table 1. We use bold font where we use a hyperparameter different from Sodhani et al. [55]. Similarly to Sodhani et al. [55], we use the v1 version of Metaworld for our experiments[2]. Sodhani et al. [55] use a shared entropy loss weight $\alpha$ for PCGrad and separate $\alpha$ for unitary scalarization[3]. In our experiments, use shared $\alpha$ for all of the methods for fairness. Since it is a single number (rather than a vector), we used unitary scalarization to update $\alpha$ for all SMTOs apart from PCGrad which was already implemented in [55].

We use the same network architecture as in Sodhani et al. [55], i.e. a three-layered feedforward fully-connected network with 400 hidden units per layer for both, the actor and the critic. The actor is shared across all tasks as well as the critic.

To normalize rewards, we keep track of first and second moments in the buffer and normalise the rewards by their standard deviation: $r'_i = r_i/\hat{\sigma}_i$, where $\hat{\sigma}_i$ is the sample standard deviation of the rewards for environment $i$.

Sodhani et al. [55] average the gradient for unitary scalarisation and pcgrad, whereas our SMTO implementations sum the gradients, i.e. effectively using larger learning rates (apart from MGDA that assures that all the aggregation weights sum to 1). We tried reducing the learning rate for SMTOs that sum (RLW Norm., RLW Diri., and GradDrop) both for MT10 and MT50, but it worked worse for these methods and we kept the default learning rate for them as well. We had to use a smaller learning rate for IMTL, because with the default one it crashed at the beginning of training due to numerical overflow. Smaller learning rate did not prevent it from crashing, but this happened much later.

We tried $10^6$, $2 \times 10^6$, and $4 \times 10^6$ for the replay buffer size with the last being superior in terms of stability. Additionally, for $l_2$ actor regularization, we tried 0.0001 and 0.0003 with the latter being slightly superior for the baseline. We tried the same options for other SMTOs, and picked the best option for each of the method. For MGDA, no regularisation works best, most likely due to a strong regularization effect of the method itself, which is mirrored by our supervised learning results. PCGrad and Graddrop work best with the regularization coefficient of 0.0001. Both RLW variants use the same coefficient as the baseline (0.0003).

For MT50, we took the best MT10 hyperparameters, and we believe one could obtain even better results for unitary scalarisation since it is much faster to tune compared to other SMTOs (e.g. 15 hours for unitary scalarisation vs 9 days for PCGrad).

## C.3 Software Acknowledgments and Licenses

Our codebase is built upon several prior works: [54], [43], [40] and [55]: all of them were released under a MIT license. We also acknowledge Tseng [59], upon which we built some of our code. Multi-MNIST is based on MNIST dataset that is released under Creative Commons Attribution-Share Alike 3.0 license. The code for generating Multi-MNIST dataset was taken from Sener and Koltun [54] released under MIT license. CelebA dataset has a custom license allowing non-commercial research purposes. More details can be found on the project website:`http://mmlab.ie.cuhk.edu.hk/projects/CelebA.html`. Cityscapes also has a custom license allowing non-commercial research purposes. The full text of the license can be found on the project website:`https://www.cityscapes-dataset.com/license/`. Metaworld, used for RL experiments is released under MIT license.

---

[2]`https://github.com/rlworkgroup/metaworld.git@af8417bfc82a3e249b4b02156518d775f29eb289`
[3]`https://mtrl.readthedocs.io/en/latest/pages/tutorials/baseline.html`

# D Supplementary Supervised Learning Experiments

Table 1: Hyperparameters of the RL experiments. Hyperparameters different from Sodhani et al. [55] are in bold.

| Hyperparameter | Value |
|---|---|
| **All methods** | |
| – training steps | 2,000,000 |
| – batch size | 1280 |
| – **Replay buffer size** | **4,000,000** |
| – actor learning rate | 0.0003 |
| – critic learning rate | 0.0003 |
| – entropy $\alpha$ learning rate | 0.0003 |
| – shared entropy $\alpha$ | True |
| – runs | 10 |
| – discounting $\gamma$ | 0.99 |
| **Unit. Scal.** | |
| – **actor $l_2$ coeff.** | **0.0003** |
| **PCGrad** | |
| – **actor $l_2$ coeff.** | **0.0001** |
| **RLW Norm.** | |
| – normal mean | 0 |
| – normal std | 1 |
| – actor $l_2$ coeff. | 0.0003 |
| **RLW Diri.** | |
| – $\alpha$ | 1 |
| – actor $l_2$ coeff. | 0.0003 |
| **GradDrop** | |
| – k | 1 |
| – p | 0.5 |
| – actor $l_2$ coeff. | 0.0001 |
| **MGDA** | |
| – gradient normalization | $L_2$ |
| – actor $l_2$ coeff. | 0.0 |
| **IMTL** | |
| – actor learning rate | 0.00003 |
| – critic learning rate | 0.00003 |
| – entropy $\alpha$ learning rate | 0.00003 |
| – actor $l_2$ coeff. | 0.0 |

This section presents supervised learning results omitted from §4.1. In particular, we show additional plots for the experiments of §4.1, then present an analysis of the regularising effect of SMTOs in the absence of single-task regularization (§D.2), and conclude with an ablation study on GradDrop's dependency on the sign of per-task gradients (§D.3).

## D.1 Addendum

This section complements the plots presented in §4.1. In particular, we show the test and runtime results in table form, along with the behavior of the validation metrics and of the training loss over the training epochs. Plots for Multi-MNIST, CelebA, and Cityscapes are reported in Figures 7, 8 and 17, respectively. The Multi-MNIST plots show that all optimizers are relatively stable and drive each task's loss towards its global minimum of 0. The behavior of the CelebA training loss demonstrates heavier regularization (compare with the unregularized plot in Figure 9(a)). Except IMTL and MGDA, for which the tuned values of the weight decay prevent overfitting, the other optimizers display very similar validation and training curves, and start overfitting around epoch 30. Considering that most SMTOs required less regularization (see §4.1.2), the results are consistent with our interpretation of SMTOs as regularizers in §5. The Cityscapes plots display a certain instability across training epochs, as demonstrated by the various peaks and valleys in the metrics. Nevertheless, in spite of a factor 10 difference in scale, both training losses are rapidly driven towards 0 by most optimizers.

## D.2 Unregularized Experiments

Figures 5, 9(a) and 9(b) respectively report the average task validation accuracy, the multi-task training loss, and the multi-task validation loss at each training epoch. The regularizing effect of SMTOs compared to unitary scalarization is shown by: (i) the delay of the onset of overfitting on the validation data in figure 5, (ii) the reduction of the convergence rate on the training loss in figure 9(a) (compare with figure 8(b)), and (iii) the fact that validation and training losses remain positively correlated for larger numbers of epochs. In fact, the behavior of both the training and validation loss for the SMTOs closely parallels that of $\ell_2$-regularized unitary scalarization, with differing degrees of regularization. We further note that unregularized IMTL displays a certain instability (compare with the regularized version in figure 8(a)).

The addition of dropout layers further reduces overfitting, improves stability (reduced confidence intervals) and pushes the average validation curve upwards, motivating its use on all optimizers for the

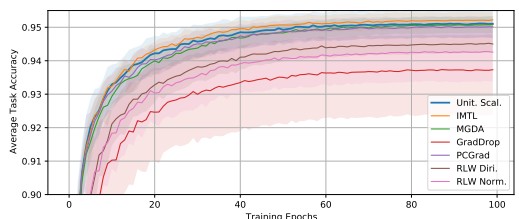

(a) Mean (and 95% CI) average task validation accuracy per training epoch.

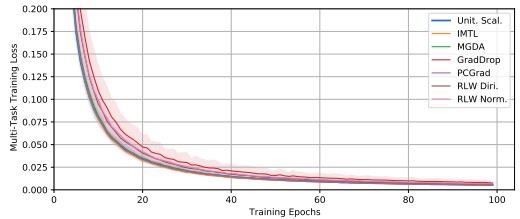

(b) Mean (and 95% CI) training multi-task loss $\mathcal{L}^{\mathrm{MT}}$ per epoch.

| MTO | Average Task Accuracy | Epoch Runtime [s] |
|---|---|---|
| Unit. Scal. | 9.476e-01 $\pm$ 4.368e-03 | [3.510e+00, 3.617e+00] |
| IMTL | 9.487e-01 $\pm$ 2.533e-03 | [3.695e+00, 3.996e+00] |
| MGDA | 9.478e-01 $\pm$ 1.977e-03 | [3.491e+00, 3.617e+00] |
| GradDrop | 9.347e-01 $\pm$ 1.282e-02 | [3.508e+00, 3.589e+00] |
| PCGrad | 9.479e-01 $\pm$ 3.578e-03 | [3.807e+00, 3.928e+00] |
| RLW Diri. | 9.430e-01 $\pm$ 2.973e-03 | [3.790e+00, 4.005e+00] |
| RLW Norm. | 9.399e-01 $\pm$ 8.929e-03 | [3.894e+00, 4.225e+00] |

(c) Mean and 95% CI of the avg. task test accuracy across runs, and interquartile range for the training time per epoch.

Figure 7: Additional figures for the comparison of various SMTOs with the unitary scalarization on the MultiMNIST dataset [54].

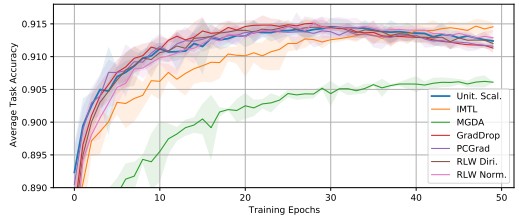

(a) Mean (and 95% CI) average task validation accuracy per training epoch.

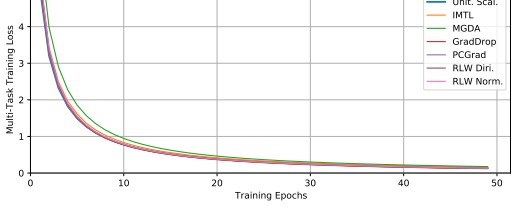

(b) Mean (and 95% CI) training multi-task loss $\mathcal{L}^{\mathrm{MT}}$ per epoch.

| MTO | Average Task Accuracy | Epoch Runtime [s] |
|---|---|---|
| Unit. Scal. | 9.090e-01 $\pm$ 7.568e-04 | [2.869e+02, 2.878e+02] |
| IMTL | 9.093e-01 $\pm$ 7.631e-04 | [3.600e+02, 3.621e+02] |
| MGDA | 9.022e-01 $\pm$ 9.687e-04 | [6.859e+02, 7.194e+02] |
| GradDrop | 9.098e-01 $\pm$ 3.383e-04 | [3.001e+02, 3.008e+02] |
| PCGrad | 9.093e-01 $\pm$ 1.108e-03 | [1.015e+04, 1.016e+04] |
| RLW Diri. | 9.099e-01 $\pm$ 7.845e-04 | [3.040e+02, 3.054e+02] |
| RLW Norm. | 9.095e-01 $\pm$ 1.012e-03 | [3.028e+02, 3.037e+02] |

(c) Mean and 95% CI of the avg. task test accuracy across runs, and interquartile range for the training time per epoch.

Figure 8: Additional figures for the comparison of various SMTOs with the unitary scalarization on the CelebA [44] dataset.

experiments of §4.1.2. Nevertheless, confidence intervals in Figure 5 still overlap due to the instability of the unregularized unitary scalarization. Figure 11 provides a more detailed comparison over 20 repetitions, confirming that the combined use of dropout layers and $\ell_2$ regularization improves average performance and reduces the empirical variance for unitary scalarization. Furthermore, Figure 10 shows that regularization improves the peak average validation performance for all algorithms, demonstrating the need of tuning $\lambda$ also for SMTOs. We conclude by pointing out that even without regularization, when carefully tuned, the maximal performance over epochs of unitary scalarization is comparable to SMTOs in Figure 5.

### D.3   Sign-Agnostic GradDrop

We will now present an ablation study on GradDrop, investigating the effect of the sign of per-task gradients on the SMTO's performance. Specifically, we compare the performance of GradDrop

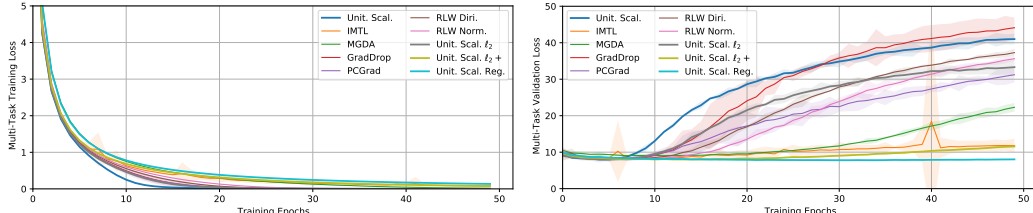

(a) Mean and 95% CI (3 runs) multi-task training loss per epoch.

(b) Mean and 95% CI (3 runs) multi-task validation loss per training epoch.

Figure 9: Additional figures for the unregularized comparison of various SMTOs with the unitary scalarization on CelebA. SMTOs provide varying degrees of regularization.

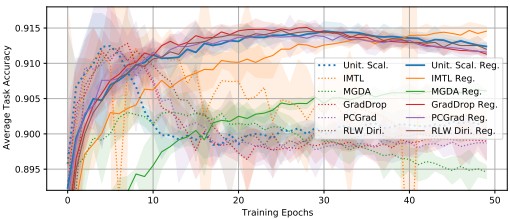

Figure 10: Effect of regularization (dropout layers and weight decay) on the average task validation accuracy for all considered optimizers on the CelebA dataset: regularization improves the average performance of all algorithms.

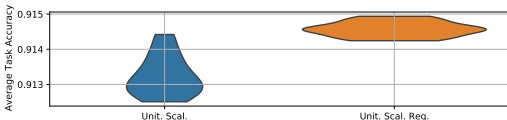

Figure 11: Effect of regularization (dropout layers and weight decay) on unitary scalarization on the CelebA dataset: violin plots (20 runs) for the best avg. task validation accuracy over epochs. The width at a given value represents the proportion of runs yielding that result. Regularization improves the average performance while decreasing its variability.

with a sign-agnostic version of its stochastic gradient masking (which we refer to as "Sign-Agnostic GradDrop"), whose update direction is defined as follows:

$$\mathbf{g} = -\left(\frac{\partial \mathbf{z}}{\partial \boldsymbol{\theta}_\|}\right)^T \left(\sum_{i \in \mathcal{T}} \mathbf{u}_i \odot \nabla_{\mathbf{z}} \mathcal{L}_i\right),$$

where $\mathbf{u}_i, \nabla_{\mathbf{z}} \mathcal{L}_i \in \mathbb{R}^{n \times r}$ and, for all $i \in \mathcal{T}$, $\mathbf{u}_i$ is i.i.d. according to $\mathbf{u}_i[j,k] \sim \text{Bernoulli}(p) \; \forall j \in \{1, \ldots, n\}, k \in \{1, \ldots, r\}$. Differently from a similar study carried out by Chen et al. [11], we tuned the hyper-parameter of the sign-agnostic masking in the following range: $p \in \{0.1, 0.25, 0.5, 0.75, 0.9\}$.

The experimental setup complies with the one described in appendix C.1. Figure 12, plotting test and validation results for the CelebA dataset [44], shows that the performance of Sign-Agnostic GradDrop closely matches the original algorithm. Therefore, sign conflicts across per-task gradients do not seem to play a significant role in GradDrop's performance.

# E  Supplementary Reinforcement Learning Experiments

## E.1  Addendum

This section presents additional plots for the RL experiments in §4.2. Specifically, Figure 13 re-plots Figure 4(a) and 4(b) with the omitted IMTL results, while Figure 14 shows the learning curves omitted

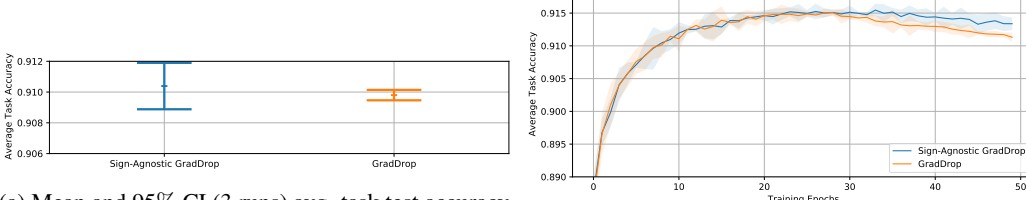

(a) Mean and 95% CI (3 runs) avg. task test accuracy.

(b) Mean and 95% CI (3 runs) avg. task validation accuracy per training epoch.

Figure 12: Comparison of GradDrop [11] with sign-agnostic masking of the shared-representation gradients on the CelebA dataset [44]. No statistically relevant difference between the two methods can be observed for the majority of the epochs.

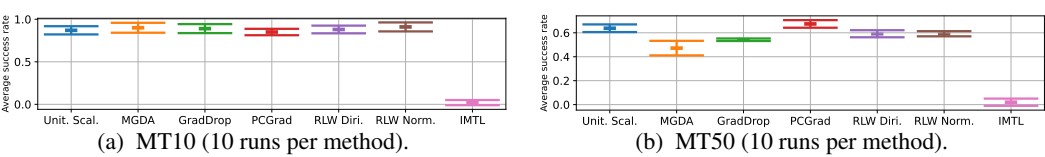

(a) MT10 (10 runs per method).      (b) MT50 (10 runs per method).

Figure 13: Mean and 95% CI for the best avg. success rate on Metaworld. None of the SMTOs significantly outperforms unitary scalarization.

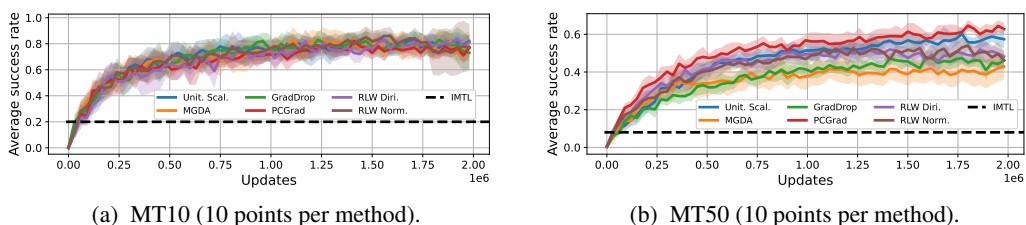

(a) MT10 (10 points per method).      (b) MT50 (10 points per method).

Figure 14: Mean and 95% CI for the avg. success rate on Metaworld. None of the SMTOs significantly outperforms unitary scalarization.

from §4.2. As pointed out in §4.2, none of the IMTL runs successfully terminated due to numerical instability. Indeed, Liu et al. [42] show that, in supervised settings, coefficients do not fluctuate much across epochs [42, Figure 4, appendix B] and never become negative. By contrast, up to 50% of the scaling coefficients $\alpha$ are negative in our experiments, thus reversing subtask gradient directions. MGDA, which constrains the weights, is more stable and is comparable to unitary scalarization. In order to avoid incomplete curves and unfair calculations of the mean, Figure 14 plots the highest value ever achieved by *any* seed as a dashed horizontal line. The IMTL results in Figure 13, instead, report the best average success rate of each seed until its termination.

## E.2 Ablation studies

Figure 18 presents our ablations for MT10 experiments. Due to computational constraints, we ran ablations on the unitary scalarization and PCGrad since these are the two methods previously tested in the RL setting.

Figure 15 shows ablation studies on the effect of regularization on MT10 and MT50. In spite of CI overlaps, actor $l_2$ regularization pushes the average higher on both benchmarks, motivating our use of regularization for the experiments in §4.2. Furthermore, the gap between the averages tends to widen with the number of updates on MT50, suggesting improved stabilization.

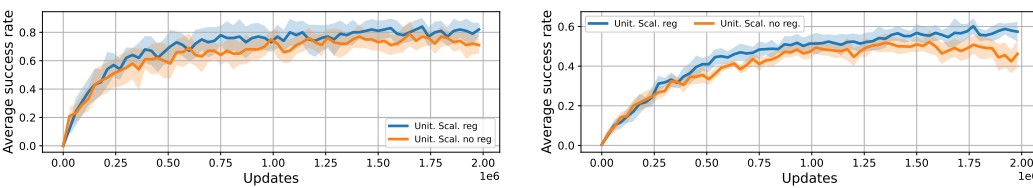

(a) MT10 average performance (10 runs) and 95% CI.   (b) MT50 average performance (5 runs) and 95% CI.

Figure 15: For both MT10 and MT50, actor $l_2$ regularization pushes the average higher for unitary scalarization.

## E.3 Sensitivity to Reward Normalization

Figure 16 shows that multitask agent performance is highly sensitive to the reward normalization moving average hyperparameter[4] motivating our buffer normalization in Section 4.2.

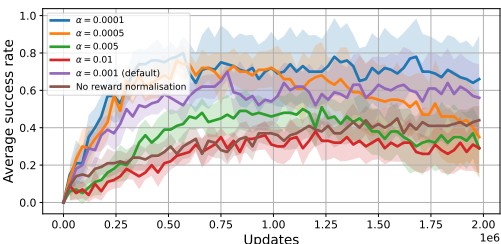

Figure 16: The learning outcomes of a Multitask SAC agent vary considerably depending on the reward normalisation hyperparameter. Each of the curves represents and average of 10 runs with shaded 95% confidence interval.

---

[4] https://github.com/facebookresearch/mtenv/blob/4a6d9d6fdfb321f1b51f890ef36b5161359e972d/mtenv/envs/metaworld/wrappers/normalized_env.py#L69

(a) Mean and 95% CI of the test metrics across runs, and interquartile range for the training time per epoch.

| MTO | Absolute Depth Error | Relative Depth Error | Segmentation Accuracy | Segmentation mIOU | Epoch Runtime [s] |
|---|---|---|---|---|---|
| Unit. Scal. | $1.301\text{e-}02 \pm 2.342\text{e-}04$ | $4.761\text{e+}01 \pm 5.148\text{e+}00$ | $9.196\text{e-}01 \pm 2.913\text{e-}04$ | $7.012\text{e-}01 \pm 6.001\text{e-}04$ | [3.228e+02, 3.241e+02] |
| IMTL | $1.281\text{e-}02 \pm 7.521\text{e-}04$ | $4.389\text{e+}01 \pm 6.984\text{e-}01$ | $9.164\text{e-}01 \pm 2.828\text{e-}03$ | $6.967\text{e-}01 \pm 4.785\text{e-}03$ | [7.329e+02, 7.373e+02] |
| MGDA | $1.418\text{e-}02 \pm 2.331\text{e-}04$ | $4.750\text{e+}01 \pm 1.466\text{e+}01$ | $9.189\text{e-}01 \pm 2.636\text{e-}04$ | $6.999\text{e-}01 \pm 3.124\text{e-}03$ | [7.251e+02, 7.269e+02] |
| GradDrop | $1.293\text{e-}02 \pm 2.757\text{e-}04$ | $4.674\text{e+}01 \pm 7.709\text{e+}00$ | $9.193\text{e-}01 \pm 1.282\text{e-}03$ | $7.024\text{e-}01 \pm 3.628\text{e-}03$ | [5.196e+02, 5.215e+02] |
| PCGrad | $1.294\text{e-}02 \pm 2.284\text{e-}04$ | $4.380\text{e+}01 \pm 5.165\text{e+}00$ | $9.198\text{e-}01 \pm 9.119\text{e-}04$ | $7.025\text{e-}01 \pm 6.531\text{e-}04$ | [4.202e+02, 4.212e+02] |
| RLW Diri. | $1.305\text{e-}02 \pm 4.155\text{e-}04$ | $4.810\text{e+}01 \pm 2.259\text{e+}00$ | $9.199\text{e-}01 \pm 1.247\text{e-}03$ | $7.037\text{e-}01 \pm 1.989\text{e-}03$ | [3.161e+02, 3.164e+02] |
| RLW Norm. | $1.301\text{e-}02 \pm 5.528\text{e-}04$ | $4.630\text{e+}01 \pm 2.751\text{e+}00$ | $9.192\text{e-}01 \pm 4.962\text{e-}04$ | $7.006\text{e-}01 \pm 4.580\text{e-}03$ | [3.194e+02, 3.210e+02] |

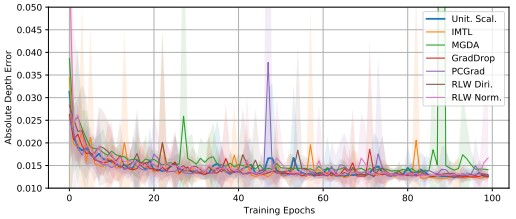

(b) Mean (and 95% CI) absolute depth validation error per training epoch.

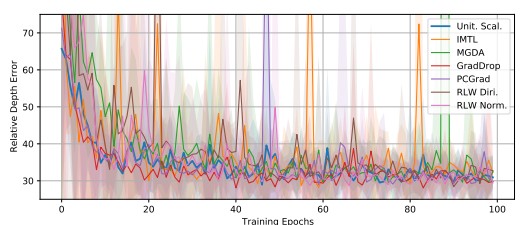

(c) Mean (and 95% CI) relative depth validation error per training epoch.

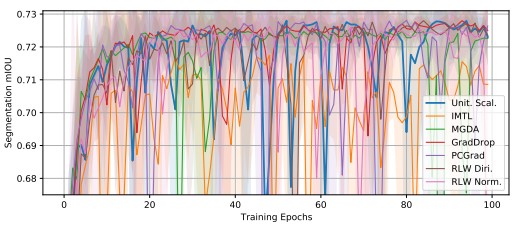

(d) Mean (and 95% CI) validation segmentation mIOU per training epoch.

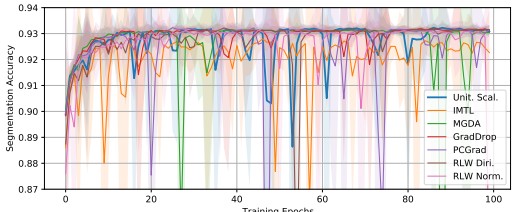

(e) Mean (and 95% CI) validation segmentation accuracy per training epoch.

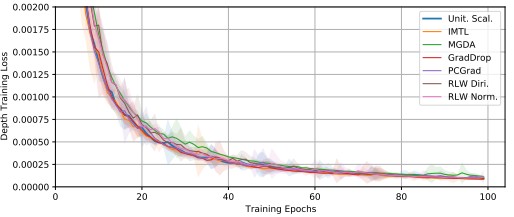

(f) Mean (and 95% CI) training depth loss per epoch.

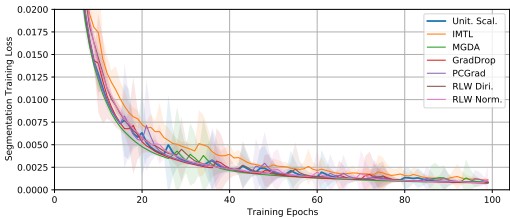

(g) Mean (and 95% CI) training segmentation loss per epoch.

Figure 17: Additional figures for the comparison of SMTOs with the unitary scalarization on the Cityscapes [13] dataset.

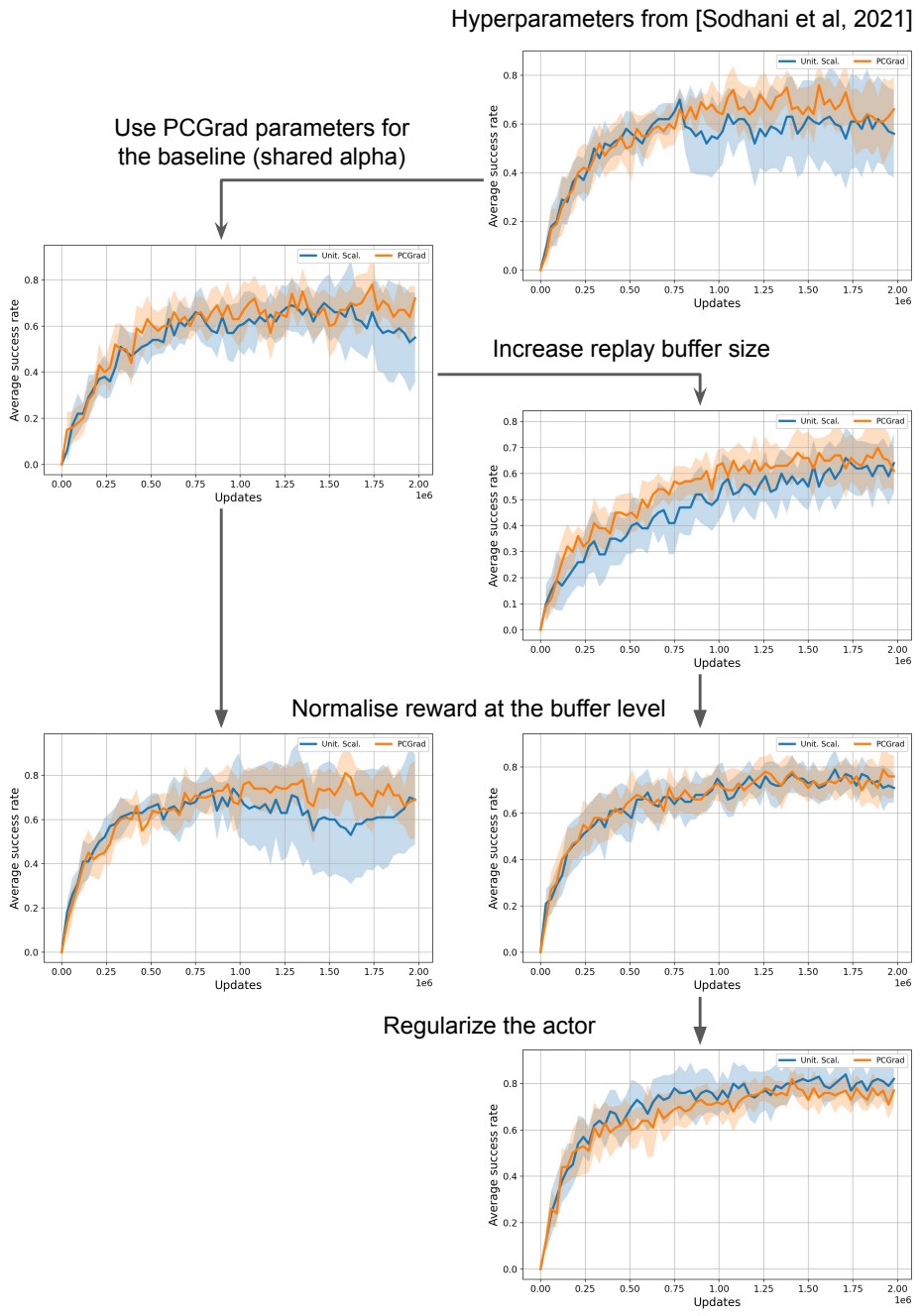

Figure 18: Metaworld's MT10 ablation experiments.