# OpenReview forum: "In Defense of the Unitary Scalarization for Deep Multi-Task Learning"
_NeurIPS.cc/2022/Conference — NeurIPS 2022 Accept_

### Official Review · Reviewer_9gYq · 2022-07-07

**Rating:** 5
**Confidence:** 4
**Soundness:** 3 good
**Presentation:** 3 good
**Contribution:** 3 good

**Summary:**

This paper considers the multitask learning setup (MTL) in which a model is trained to perform several tasks. Several existing MTL approaches argue against the simple baseline of using the sum per-task losses (unitary scalarization) as the training objective. Instead, SOTA MTL approaches usually manipulate the per-task gradients to obtain a joint update direction. Thus, the more complex per-task methods increase the training time and implementation complexity. The authors show that by adding proper regularization and carefully tuning the unitary scalarization baseline, it can perform on-par with existing MTL approaches. The paper raises some valid concerns regarding the evaluation and experimental methodology in the MTL literature. However, some questions about the experimental setup and missing comparisons/benchmarks impair the authors' claim.


**Questions:**

See above.

**Strengths And Weaknesses:**

**Strengths:**
- The paper is well organized and well written.
- This paper offers a novel view of SOTA MTL methods as regularization methods.
- The authors perform an important evaluation of recent MTL methods and raise valid concerns regarding the evaluation and experimental methodology in the MTL literature.
- Novel analysis and view of well-known MTL methods through the lens of regularization.

**Weaknesses:**
- Most MTL approach uses the feature gradients as a surgent to the full task gradient (or uses other approaches to scale the computation), hence resulting in only a small increase of runtime.
- The arguments for presenting MTL methods as regularizers appear quite weak. While interesting and novel, it appears to provide only a partial explanation.
- The experimental setup lacks important comparisons and evaluation on more relevant benchmarks:
  - First, the authors fail to (cite [1], and) compare with recent strong MTL approaches [1,2,3] that show large performance gains w.r.t unitary scalarization.
  - The results presented in [1,2] show better performance on the Cityscapes benchmark for all methods (for example in terms of mIoU). This might suggest that all methods are under-optimized.
  - A better metric for MTL evaluation is the relative task improvement (compared to STL), used in previous MTL literature [1,2,3].
  - For Cityspcaes, it’s not clear how the early stopping was performed. How do you reduce the different metrics into a single metric as the early stopping criterion? Metrics have different scales and meanings. A good approach is using the relative task improvement as the stopping criterion.
  - The supervised learning benchmarks are insufficient: (i) Multi-MNIST is a toy dataset with only two tasks; (ii) The vast majority of methods quickly overfit the Celeb-A benchmark. Hence, this benchmark is not the best choice for arguing something concerning MTL optimization: the effective optimization process ends early, so it’s hard to learn something meaningful about the optimization dynamics. (iii) Finally, the Cityspcaes dataset consists of only two tasks that were shown to benefit each other in previous works. Hence, we cannot learn from it about optimization with a large number of possibly conflicting tasks.
  - I suggest the authors add more relevant and challenging MTL benchmarks like the QM9 with 11 tasks (used in [1]) and the NYUv2 dataset with 3 tasks (commonly used through MTL lit.).

References:
- [1] Multi-Task Learning as a Bargaining Game. ICML 2022.
- [2] Conflict-Averse Gradient Descent for Multi-task learning. NeurIPS 2021.
- [3] RotoGrad: Gradient Homogenization in Multitask Learning. ICLR 2022.

---

> ### Author Response · Authors · 2022-08-02
> **Response to reviewer 9gYq**
>
> > Most MTL approach uses the feature gradients as a surgent to the full task gradient [...] hence resulting in only a small increase of runtime.
>
> As shown by the reported runtimes (for instance, Figure 3e), the overhead might be non-negligible in spite of the reliance on per-task feature gradients. The overhead is even more significant (Figures 4c, 4d) when the architecture is not encoder-decoder (so no feature gradients are available).
> Furthermore, popular SMTOs imply additional implementation complexity, which does not appear to result in a corresponding performance increase.
>
> > The arguments for presenting MTL methods as regularizers appear quite weak. While interesting and novel, it appears to provide only a partial explanation.
>
> As we mention in the conclusions (lines 386-388), we agree that our arguments provide only a partial explanation. We hope that Section 5 will catalyze future work on the topic. Given the complexity of the problem, we doubt that there is a single complete explanation. Nevertheless, we believe the empirical evidence presented in section 5.2, appendix D.2, and the new experiment in appendix D.3 to sufficiently support our arguments, which we believe to be worth sharing with the wider community.
>
> > [...] the authors fail to (cite [1], and) compare with recent strong MTL approaches [1,2,3] that show large performance gains w.r.t unitary scalarization.
>
> As we mention in the paper and general response, our goal is not to outperform all SMTOs, but to challenge the efficacy of a popular line of previous work, providing an alternative interpretation. [1,2,3] are recent works and have not yet encountered widespread adoption: [1] was published even after the NeurIPS deadline (ICML 2022). Nevertheless, we included it in the related work and mentioned it in section 5.2, as it shares MGDA’s convergence set. [2] displays a large runtime (only 2x and 5x speed up compared to PCGrad on MT10 and MT50), lacks substantial improvements over the SMTOs we considered (excluding RL results, which are weak compared ours in the same CARE setup), and introduces an additional hyper-parameter (c, grid-searched over 9 values without a separate validation set in Cityscapes/NYUv2). [3] displays comparable performance to unitary scalarization in their own CelebA experiments, displays a cubic time and quadratic memory dependency on the size of the representation (would not be applicable to our Cityscapes experiment), and it only supports encoder-decoder architectures (so is not applicable to our RL experiments). When an additional page is allowed, we will add this discussion to the paper.
>
> > The results presented in [1,2] show better performance on the Cityscapes benchmark for all methods (for example in terms of mIoU). This might suggest that all methods are under-optimized.
>
> [1,2] use a SegNet + MTAN, whereas, matching the setup from RLW, we use a ResNet-50 + ASPP predictive heads, see Appendix C.1.3 for more details. Our goal is not to outperform SMTOs, but provide a fair ground for comparison. We added a sentence to Appendix C.1.3. mentioning that better Cityscapes performance might be achieved when using MTAN.
>
> > A better metric for MTL evaluation is the relative task improvement (compared to STL), used in previous MTL literature [1,2,3].
>
> All the problems we consider, except Cityscapes, have homogeneous metrics per task and are naturally compared in terms of average performance. On Cityscapes, we believe the relative task improvement to be too sensitive to a single metric (e.g., the relative depth error in table 3 of [1]).
>
> > For Cityscapes, it’s not clear how the early stopping was performed. How do you reduce the different metrics into a single metric as the early stopping criterion?
>
> As detailed in lines 227-229, the model employed at test time for each metric is defined as the model with the best validation result for that metric across epochs.
>
> > The supervised learning benchmarks are insufficient
>
> As in all previous works, we treat MNIST as a sanity check. We disagree that the overfitting on CelebA makes it unsuitable for assessing SMTOs: quite the opposite. It is fairly interesting to see how MTL optimizers behave on a benchmark where over-optimizing is a problem, and where it is routinely claimed that SMTOs beat unitary scalarization. Such a benchmark provides a lot of information on the optimization dynamics of an MTL optimizer. Finally, Cityscapes is a widely-used benchmark in the community, and its results were employed to motivate many SMTOs.
>
> > I suggest the authors add more relevant and challenging MTL benchmarks
>
> We believe that the RL experiments provide a more challenging (10 and 50 tasks) and qualitatively different setting. Indeed, previous works (see PCGrad) showed very poor unitary scalarization (and STL) results on MT10 and MT50. Furthermore, the RL setting does not employ an encoder-decoder architecture, therefore complementing the supervised learning setup.

---

> > ### Comment · Reviewer_9gYq · 2022-08-06
> > **Comments to the rebuttal**
> >
> > I would like to thank the authors for their response. The authors have made attempts to answer all my questions, however, I still have some concerns and follow-ups.
> >
> > 1. As noted by the authors, the narrative of our paper is primarily an experimental one. As such, I still believe the supervised learning results are insufficient, due to the reasons stated in my original comment.
> > 2.
> > > "... the model employed at test time for each metric is defined as the model with the best validation result for that metric across epochs."
> > Does this mean that you potentially evaluate two different models (each at different optimization stages) rather than one MTL model? If so, this evaluation is somewhat flawed in my view.

---

> > > ### Author Response · Authors · 2022-08-06
> > > **response**
> > >
> > > We sincerely thank the reviewer for following up and explaining his concerns.
> > >
> > > 1. Considering their complexity and overhead, the final motivation behind the adoption of the most popular SMTOs (such as PCGrad, MGDA, GradDrop, IMTL)
> > > was entirely experimental, and based on performance on widespread benchmarks such as Multi-MNIST (PCGrad, MGDA),
> > > CelebA (MGDA, PCGrad, IMTL, GradDrop, RLW), and Cityscapes (MGDA, PCGrad, IMTL, RLW).
> > > Given their role in the original justification of these SMTOs, we believe these supervised benchmarks should serve as the primary ground for our experimental analysis, and suffice to cast doubts on the effectiveness of these methods.
> > > Therefore, we do not believe that additional (and less commonly adopted, such as QM9) supervised benchmarks would particularly strengthen our narrative.
> > > Finally, we stress again that, given the different setup and the inherent and well-known difficulty of the benchmark,
> > > the RL experiments on MT10 and MT50 (which were in turn employed to justify the effectiveness of PCGrad) already fulfil the request for "more relevant and challenging" benchmarks.
> > > Indeed, MT10 and MT50 both have a large number of tasks (10 and 50, respectively), and the "effective optimization process" does not end early, as shown in Figure 14.
> > >
> > > 2. We respectfully disagree that our methodology implies that the evaluation is flawed.
> > > As we employ the same methodology consistently across methods, we do not have any reason to believe that it would favour any given optimizer over the others.
> > > It is hence fair, which is the main goal of our evaluation.
> > > Furthermore, we argue that there is no single correct way to report results for heterogeneous MTL benchmarks:
> > > - the relative task improvement assumes that the delta in each metric's improvement is to be weighted linearly: is this a reasonable assumption?
> > > - given that multi-task training improves on the final performance of each single task, the practitioner might want to prioritize a single task's performance over the others;
> > > - even without a task/metric preference, we do not see why only a single model checkpoint should be stored.
> > > - as highlighted by our Figure 17, all Cityscapes validation metrics display a "spiky" behaviour over the epochs: the use of a single checkpoint would imply failing to capture many of the performance peaks;
> > > - most of the considered methods that previously reported Cityscapes results (MGDA, PCGrad, IMTL), do not provide details on their model selection methodology, and report validation results, so it's hard to argue that there is an established common practice on this;

---

> > > > ### Comment · Reviewer_9gYq · 2022-08-07
> > > > **Response to authors**
> > > >
> > > > I thank the authors for their reply.
> > > > 1. I agree with the authors that the presented experiments suffice to cast doubts on the effectiveness of SMTOs. However, I still feel that the supervised learning results are partial and not strong enough.
> > > > 2. One of the main motivations for using MTL is to reduce computational demand at inference time (as a result of using a shared backbone for all tasks). Thus, evaluating a single MTL model, rather than different STL ones makes more sense.

---

> > > > > ### Author Response · Authors · 2022-08-08
> > > > > **response**
> > > > >
> > > > > We thank the reviewer for the response.
> > > > >
> > > > > > I agree with the authors that the presented experiments suffice to cast doubts on the effectiveness of SMTOs. However, I still feel that the supervised learning results are partial and not strong enough.
> > > > >
> > > > > We are extremely glad that the reviewer finds that "the presented experiments suffice to cast doubts on the effectiveness of SMTOs". Indeed, as expressed in the paper and the discussion above, this is exactly the goal of our work. We do not aim to claim that unitary scalarization will systematically outperform all SMTOs, and we agree that such a claim would necessitate an even more comprehensive experimental comparison.
> > > > >
> > > > > > One of the main motivations for using MTL is to reduce computational demand at inference time. Thus, evaluating a single MTL model, rather than different STL ones makes more sense.
> > > > >
> > > > > As we mentioned in the previous response, we aim to evaluate all optimizers on a fair ground, and we believe that this is achieved through the consistent use of the same evaluation procedure.
> > > > > We would be happy to open a discussion if the reviewer believes otherwise (i.e., that our Cityscapes pipeline favors any of the optimizers).
> > > > > Nevertheless, we would like to stress again the lack of a widely accepted and flawless evaluation procedure for heterogeneous MTL problems.

---

> > > > > > ### Comment · Reviewer_9gYq · 2022-08-08
> > > > > > **response to authors**
> > > > > >
> > > > > > I do not think the evaluation pipeline favors one of the optimizers; However, I believe the current evaluation does not measure what you wish to measure, namely MTL performance (i.e., the performance of one MTL model).
> > > > > > However, this is a minor comment in my review. Making the paragraph discussing Cityscape's result clearer, in terms of the evaluation process and its limitations, should address my concern.

---

> > > > > > > ### Author Response · Authors · 2022-08-09
> > > > > > > **response to reviewer 9gYq**
> > > > > > >
> > > > > > > Thanks for the suggestion. We included the discussion on Cityscapes model selection in Appendix C.1.3, and will clarify this in the main text when more space is allowed.

---

### Official Review · Reviewer_pDqD · 2022-07-11

**Rating:** 6
**Confidence:** 4
**Soundness:** 3 good
**Presentation:** 4 excellent
**Contribution:** 3 good

**Summary:**

The authors perform an empirical evaluation of various optimizers designed specifically for multi-task learning, particularly those that utilize per-task gradients. The experiments, which are conducted on several image classification tasks and the metaworld MT10 and MT50 multi-task RL benchmarks, generally show that unitary scalarization (simply summing loss for the tasks) is a very strong baseline for multi-task learning when its regularization (l2 + dropout) is tuned, and the benefits of more sophisticated optimizers is dubious, given their computational cost. The authors interpret their findings to mean that more sophisticated multi-task learning objectives essentially perform a form of early stopping, providing some initial theoretical arguments for why this may be the case.

**Questions:**

Overall, my biggest concern with the paper is Section 5. While I can appreciate the general intuition that a larger number of convergence points may lead to early stopping-like behavior, the authors don't convincingly argue that this is indeed the case in practice. For example, the statement that PCGrad has an "enlarged convergence set" based on the points where gradients are exactly opposed seems like a stretch- this is a vanishingly small set of points. Nonetheless, PCGrad performs reasonably favorably to the other baseline algorithms, and the alternative argument that the stochasticity in the objective leads to better performance is not developed in much depth.

One way to address these concerns would be to empirically show that, as their theoretical argument suggests, SMTOs do indeed converge to solutions that are far from the fixed points of unitary scalarization. i.e., at SMTO convergence, is the unitary scalarization update magnitude still large? Alternatively, can the authors compare the flatness of the minima for regularization unitary scalarization and SMTOs? The authors invoke the flatness intuition, but it is not validated.

Regarding methodological novelty, I wonder if the authors can also provide more detailed discussion of their results; currently, while the results are interesting, they are not directly actionable for the community. What do the authors recommend the MTL community do with these results? How can these insights translate into better or more efficient models? I wonder if some of the text allocated to the theoretical section might be better used for such discussion.

As a minor point, Figure 5 is quite hard to interpret, especially for colorblind readers. I would suggest using multiple line styles to make it easier to compare between unit scale and SMTO lines.

**Limitations:**

Yes.

**Strengths And Weaknesses:**

Strengths:
- Thorough evaluation of multi-task learning (MTL) methods, with experimental details well-catalogued
- Surprising empirical result
- Likely to be a useful set of reference experiments within the MTL world
- Most of the paper is very well written

Weaknesses:
- Theoretical section seems hurried, and I'm not totally convinced by the line of argument (see below)
- As the authors point out, more comprehensive hyperparameter tuning would be nice, given the paper's primary contribution is providing a large comparison
- No methodological novelty (this is not a requirement for a strong paper, but without it, the key results need to be very strong)

---

> ### Author Response · Authors · 2022-08-02
> **Response to reviewer pDqD**
>
> We thank reviewer pDqD for the positive evaluation of our work and for the useful suggestions.
>
> > As the authors point out, more comprehensive hyperparameter tuning would be nice, given the paper's primary contribution is providing a large comparison
>
> We believe our hyperparameter search is quite comprehensive under reasonable compute constraints (see Appendix C for more details): our goal is not to outperform previously known results on any of the tasks, but rather to perform a fair comparison.
> Furthermore, unitary scalarization is easier and faster to tune due to its smaller runtime.
>
> > While I can appreciate the general intuition that a larger number of convergence points may lead to early stopping-like behavior, the authors don't convincingly argue that this is indeed the case in practice. [...] One way to address these concerns would be to empirically show that, as their theoretical argument suggests, SMTOs do indeed converge to solutions that are far from the fixed points of unitary scalarization.
>
> We thank the reviewer for the suggestion, which we considered by adding a new experiment that tracks the magnitude of the sum of the gradients (the unitary scalarization update) across SMTO epochs. The results are presented in appendix D.3. As mentioned in the response to all reviewers, the magnitude of the update towards convergence is larger for SMTOs compared to unitary scalarization. However, most of these algorithms display a smaller update magnitude than unitary scalarization in the first 10 epochs. We can therefore conclude that STMOs steer the optimization towards points that are not fixed points of unitary scalarization. Furthermore, the magnitude of the update towards convergence is noticeably larger for IMTL and MGDA, both of which were shown to have a larger convergence set in our analysis.
>
> > The alternative argument that the stochasticity in the objective leads to better performance is not developed in much depth. [...] can the authors compare the flatness of the minima for regularization unitary scalarization and SMTOs?
>
> We thank the reviewer for the suggestion, which we agree is useful.  However, due to the computational expense associated with accurate estimators of the local flatness (see https://openreview.net/pdf?id=H1oyRlYgg, equation (4)) we leave this empirical analysis for future work.
>
> > […] while the results are interesting, they are not directly actionable for the community. What do the authors recommend the MTL community do with these results? How can these insights translate into better or more efficient models?
>
> We are glad the reviewer finds our results interesting. We believe they are also actionable, along two directions:
> 1. We identify an important issue in the existing literature: underreported unitary scalarization results as well as other evaluation weaknesses (reporting validation results, using a single seed, lack of confidence intervals). This is a direct call to change the evaluation practices in the community.
> 2. The main goal of practitioners is to solve the problem at hand while minimizing the employed resources. In this sense, we advise them to try unitary scalarization before testing SMTOs.
>
> > As a minor point, Figure 5 is quite hard to interpret, especially for colorblind readers. I would suggest using multiple line styles to make it easier to compare between unit scale and SMTO lines.
>
> Thank you for your suggestion. We updated Figure 5 using the colorblind-friendly scheme from Seaborn and multiple line styles as well.

---

> > ### Comment · Reviewer_pDqD · 2022-08-10
> > **Response to author response**
> >
> > I appreciate the authors' response. It's a fair point that unitary scalarization's ease of tuning can be considered a virtue of the method, and that the results may be immediately actionable to practitioners. I'm also glad to see Figure 5 updated for improved accessibility!
> >
> > However, regarding the authors' argument that because gradient magnitudes for SMTOs and unitary scalarization are large and small in different points during training, "we can therefore conclude that SMTOs steer the optimization towards points that are not fixed points of unitary scalarization," I'm not convinced this is a valid conclusion. For example, the curves $y=x^2$ and $y=x^4$ will have the same fixed points, but different gradient magnitudes in different regimes of closeness to the optimum. In addition, I'm not convinced that estimating local flatness is computationally infeasible, especially for some of the smaller networks considered in this paper.
> >
> > Overall, I think the authors did a reasonable job of addressing some of my concerns, but not sufficiently to warrant increasing my score.

---

### Official Review · Reviewer_Ey2v · 2022-07-11

**Rating:** 6
**Confidence:** 3
**Soundness:** 3 good
**Presentation:** 3 good
**Contribution:** 3 good

**Summary:**

This work is interested in re-examining the efficacy of multi-task optimization methods, termed STMOs in the paper. The authors believe that prior STMO work does not treat the "unitary scalarization" baseline, i.e. the simple sum of objectives, fairly and that, when fairly treated, unitary scalarization can achieve comparable results to SMTOs.

The core contribution of this work is a large set of experiments on 4 multi-task datasets (3 vision and 1 reinforcement learning) in which the authors compare the distribution (mean & std.) of performance using 4 common SMTOs, 2 random weighting schemes, and unitary scalarization. Their results show that no individual method significantly outperforms unitary scalarization across all tasks. Although some methods may outperform unitary scalarization on certain tasks, unitary scalarization is always well within their standard deviation, and moreover the benefits pf SMTOs are never consistent across datasets. In addition, SMTOs have a significant effect on training time, increasing it drastically for little-to-no gain in performance.

The authors postulate that this is because most  SMTOs can actually be seen as regularizers. The second contribution of this work is a theoretical analysis of SMTOs in terms of their convergence sets - MGDA, PCGrad, and IMTL increase the convergence set of optimization (i.e. the number of solutions a method may stop at), and suggest that this increase in convergence sets can behave similar to early stopping. Overall, their work suggests that current SMTOs, whose focus is on mitigating task conflict, instead benefit multi-task learning through implicit regularization effects - an effect which can be mirrored much more simply with unitary scalarization.

**Questions:**

The CelebA experiments in 5.1 suggest that SMTOs also simply converge slower - if training continued, would they continue to overfit similar to unitary scalarization? In other words, is the early stopping exhibited in practice simply an artifact of the number of epochs the models are trained for (a fixed number of epochs for all methods) or because the multi-task methods have truly "stopped"?

**Limitations:**

yes

**Strengths And Weaknesses:**

Strengths:
- A lot of multi-task learning research has focused on conflict-mitigating methods to improve negative transfer. However, this exhaustive  set of experiments suggests that this focus has not been as beneficial to multi-task learning as previously believed. This is an important result which suggests that future MTL work should rethink how task conflict is approached.
- The number of experiments run is helpful, and is a meaningful contribution alone - it provides a reasonable reference for future multi-task work to build off of. Moreover, the authors have promised to release the code for all SMTOs.
- The analysis of SMTOs acting as regularizers is interesting - To the best of my knowledge this is the first joint theoretical treatment of the effects of SMTOs on the optimization.

Weaknesses:
- There seems to be a disconnect between the analysis of SMTOs and the empirical results of section 5.1:
	- Much of the analysis is centered around showing that there exist solutions which an SMTO may converge to, yet a unitary scalarization approach will not. To me this suggests that SMTOs will, in some senses, early-stop themselves. However, the results in 5.1 on CelebA suggest that SMTOs converge slower than unitary scalarization, and that the early stopping regularization arises because the max number of training epochs has been reached.
- At face value, the contributions of the paper seem somewhat limited - the core contribution of this paper is a large set of experiments using prior methods. I do not think this is a major weakness, because the results are novel and important. I think the paper would benefit from much more focus on section 5 - the biggest contribution of the paper, in my mind, is the connection between regularization and SMTOs which help to explain the results in section 4. However, section 5 is comparatively small and empirical analysis of this connection is very limited (5.1).

---

> ### Author Response · Authors · 2022-08-02
> **Response to reviewer Ey2v**
>
> We thank reviewer Ey2v for the detailed questions and analysis, and for the positive feedback.
>
> > At face value, the contributions of the paper seem somewhat limited [...] I think the paper would benefit from much more focus on section 5 - the biggest contribution of the paper, in my mind, is the connection between regularization and SMTOs which help to explain the results in section 4. However, section 5 is comparatively small and empirical analysis of this connection is very limited (5.1).
>
> We consider Section 4, which, using the words of reviewer 9gYq “perform[s] an important evaluation of recent MTL methods and raise[s] valid concerns” to be the most important contribution of our paper, and we are glad that you found its results to be ‘novel and important’. We think that identifying issues in the existing literature is an important contribution to the community. We hope that Section 5, which is meant to provide only a partial explanation of our empirical results, will catalyze future work on the topic. Nevertheless, section 5 is complemented by appendix B, providing proofs and additional results, and by appendix D.2, which adds to the empirical evidence presented in section 5.2. In addition, we present a new experiment in appendix D.3 to widen our empirical analysis (see general response above).
>
> > [...] there exist solutions which an SMTO may converge to, yet a unitary scalarization approach will not. To me this suggests that SMTOs will, in some senses, early-stop themselves. However, the results in 5.1 on CelebA suggest that SMTOs converge slower than unitary scalarization, and that the early stopping regularization arises because the max number of training epochs has been reached. [...] if training continued, would they continue to overfit similar to unitary scalarization? In other words, is the early stopping exhibited in practice simply an artifact of the number of epochs the models are trained for (a fixed number of epochs for all methods) or because the multi-task methods have truly "stopped"?
>
> Figure 5.1 shows that all SMTOs reach their peak performance before epoch 20, and start overfitting from then onwards. This is further clarified by Figure 8 in appendix D.2: while the training loss of SMTOs keeps on descending after epoch 20, the validation loss starts monotonically increasing for all of them. Therefore, we believe that overfitting would worsen with more epochs.
> This behavior is consistent with our analysis, as we linked the enlarged convergence set of MGDA, IMTL and PCGrad (the other considered SMTOs regularize via the added stochasticity) to under-optimization, rather than specifically to early stopping. In other words, we believe that these algorithms will steer optimization towards regions of relatively high loss values, rather than follow the trajectory of unitary scalarization and stop early on that trajectory. We believe the new results of appendix D.3 support this.

---

> > ### Comment · Reviewer_Ey2v · 2022-08-08
> > **Rebuttal Response**
> >
> > Thank you for your response!
> >
> > >We consider Section 4 [...] to be the most important contribution of our paper
> >
> > I agree that Section 4 is a meaningful stand-alone contribution - the contribution demonstrating that SMTOs fail to meaningfully outperform the baseline across multiple benchmarks warrants acceptance in my view.
> >
> > >Therefore, we believe that overfitting would worsen with more epochs.
> >
> > I also believe this to be the case! My point is only that a slower rate of convergence is not necessarily the same thing as regularization, i.e. just because SMTOs have higher training loss and lower validation loss at epoch 50 does not mean that the SMTOs will not _eventually_ reach the same training loss and validation loss as unitary scalarization (I think a similar argument holds for gradient norms in D.3). In other words, how do methods compare to each other at a point of equivalent training loss? I think such a comparison paints a more convincing picture with respect to where different methods "steer" optimization by normalizing for how fast the method is optimizing.
> > The rate of convergence is not only affected by the regions of the loss landscape that the method traverses, but by e.g. the noise of the gradient step or the norm of the gradient induced by each method.
> >
> > Regardless - I understand that it is late in the discussion period and I am not suggesting that the manuscript needs to be updated, although I would love to hear the authors thoughts. I am happy to accept that Section 5 is not meant to be a second stand-alone contribution, but rather a supplement to the main result in Section 4.

---

> > > ### Author Response · Authors · 2022-08-09
> > > **Response**
> > >
> > > We thank reviewer Ey2v for following up, for the positive comments on Section 4 ("[...] warrants acceptance in my view."), and for the clarifications.
> > >
> > > >  My point is only that a slower rate of convergence is not necessarily the same thing as regularization, [...] how do methods compare to each other at a point of equivalent training loss? I think such a comparison paints a more convincing picture with respect to where different methods "steer" optimization by normalizing for how fast the method is optimizing.
> > >
> > > We thank the reviewer for the interesting discussion on the subject!
> > > We totally agree that separating the rate of convergence from regularization is non-trivial in this context.
> > > However, we hope that the following evidence can help disambiguate the two:
> > > - The behaviour of SMTOs in Figures 5 and 9 is strikingly similar (both in terms of the train/validation loss and the metrics) to the effect of $\ell_2$ regularization on unitary scalarization.
> > > - In Figure 5 unitary scalarization attains its peak performance around epoch 5, where its loss value is relatively large (>1, Figure 9). IMTL attains its peak performance at a lower loss value, whereas MGDA never reaches 91% average validation accuracy, regardless of the loss value it attains (its training loss goes well below 1). In other words, performance is different across IMTL/MGDA/US at the same loss value, and this can be already inferred from the provided 50 epochs of training.
> > > - Figure 11 (appendix D.3) shows that the magnitude of the sum of the gradients is comparable for many SMTOs and unitary scalarization until epoch 20. This is not the case afterwards, suggesting that the optimization is heading towards a different region of the landscape.
> > >
> > > While perhaps none of the above points taken in isolation is sufficient, we hope that their conjunction could serve as reasonable evidence that the some regularization is indeed occurring.
> > > We will strive to include some of this discussion in a future revision of the paper.

---

### Author Response · Authors · 2022-08-02
**General Response**

We thank the reviewers for their detailed feedback and suggested improvements.
We are glad that reviewers found our work to be:
- **interesting**: “*an important result which suggests that future MTL work should rethink how task conflict is approached*” (Ey2v), “*Surprising empirical result*” (pDqD), “*interesting and novel*” (9gYq).
- **relevant**: “*useful set of reference experiments within the MTL world*” (pDqD), “*perform an important evaluation of recent MTL methods and raise valid concerns*” (9gYq), “*The number of experiments run is helpful, and is a meaningful contribution alone*” (Ey2v).
- **novel:** “*this is the first joint theoretical treatment of the effects of SMTOs on the optimization*” (Ey2v), “*novel view of SOTA MTL methods as regularization methods*” (9gYq).
- **thorough**: “*exhaustive set of experiments*” (Ey2v), “*thorough evaluation*” (pDqD), “*very well written*” (pDqD), “*well organized and well written*” (9gYq).

The *main goal of our work* is to show that, when following best experimental practices, a simple and fast baseline performs on par with specialized methods that are popular in the community. Given that the specialized methods are significantly more complex, both conceptually and in runtime, our paper provides a simple and actionable insight for practitioners tackling multitask learning problems: *before adapting SMTOs to the use-case at hand, or designing a new one, test whether the unitary scalarization, coupled with standard techniques from the literature, attains the target performance*.
It is not our aim to rule out that some SMTO will outperform unitary scalarization, or to frame all of them as regularizers, but rather to challenge the efficacy of a popular line of previous work on well-established benchmarks.
As we mention in the conclusion, we do not exclude the possibility that unitary scalarization will fail on some settings, or that novel algorithms will be required to solve them. The search for such settings and the corresponding algorithmic solutions are exciting avenues for future work, and we hope that our submission will inform such a line of research.

*Theoretical section (5)*. In order to address the concerns regarding section 5, we expand the analysis with **a new experiment** on the evolution of the unitary scalarization update magnitude for SMTOs, currently shown in appendix D.3 (Figure 11) and to be included in the main paper when an additional page is allowed. While starting from a smaller update magnitude than unitary scalarization, SMTOs converge towards a region where the magnitude of the sum of the gradients is significantly larger than that of unitary scalarization. As they also display significantly larger loss than unitary scalarization (Figure 8a), we can conclude that SMTOs steer optimization towards different regions, which under-optimize the loss, rather than merely slow down convergence. We thank reviewer pDqD for the suggestion.
Nevertheless, as we state in lines 278, 386-388, section 5 only offers an initial and partial attempt at the explanation of the results of section 4, and we hope to encourage further research towards their understanding. The narrative of our paper is primarily an experimental one. Indeed, to highlight this, our paper first presents empirical results (section 4) and then discusses a potential direction of theoretical analysis (section 5).


Finally, we believe the *reinforcement learning results* are an important part of the paper, which was largely neglected in the reviews. First, they involve a completely different setting from supervised learning, with an architecture that does not conform to the encoder-decoder structure. Second, Metaworld is a challenging benchmark with a large number of tasks (10 for MT10 and 50 for MT50), for which unitary scalarization results were severely underreported in the literature.

We will respond to the individual concerns of the reviewers in separate threads. All revisions to the paper are denoted by blue text.

---

### Meta-Review · Area_Chair_BwBS · 2022-08-25

**Recommendation:** Accept
**Confidence:** Certain

**Metareview:**

This paper criticizes the SMTO methods by building a single experimental pipeline and finding that none of the SMTOs consistently outperform unitary scalarization method, which is the simplest and cheapest method for multi-task learning. To explain the findings, the authors postulate that SMTOs act as regularizers and present an analysis.

Generally, I like such a kind of paper as it would provide a criticized view and corresponding evidence as well as an in-depth analysis, which let us stop and think about the real progress we have made so far, and indeed helps the long-term development of the studied area.

The reviewers generally provided supportive comments and positive overall ratings. I also make an acceptance recommendation for this paper.


**Award:**

No

---

### Decision · Program_Chairs · 2022-09-14

Accept